# *ViTaPEs*: Visuotactile Position Encodings for Cross-Modal Alignment in Multimodal Transformers

**Fotios Lygerakis***
*Chair of Cyber-Physical Systems*
*Technical University of Leoben*

**Ozan Özdenizci**
*Institute of Machine Learning and Neural Computation*
*Graz University of Technology*

**Elmar Rückert**
*Chair of Cyber-Physical Systems*
*Technical University of Leoben*

**Reviewed on OpenReview:** *https://openreview.net/forum?id=mxzzO66Zbu&noteId=D7mRrK8JwN*

## Abstract

Tactile sensing provides local essential information that is complementary to visual perception, such as texture, compliance, and force. Despite recent advances in visuotactile representation learning, challenges remain in fusing these modalities and generalizing across tasks and environments without heavy reliance on pre-trained vision-language models. Moreover, existing methods do not study positional encodings, thereby overlooking the multi-stage spatial reasoning needed to capture fine-grained visuotactile correlations. We introduce *ViTaPEs*, a transformer-based architecture for learning task-agnostic visuotactile representations from paired vision and tactile inputs. Our key idea is a two-stage positional injection: *local* (modality-specific) positional encodings are added within each stream, and a *global* positional encoding is added on the joint token sequence immediately before attention, providing a shared positional vocabulary at the stage where cross-modal interaction occurs. We make the positional injection points explicit and conduct controlled ablations that isolate their effect before a token-wise nonlinearity versus immediately before self-attention. Experiments on multiple large-scale real-world datasets show that *ViTaPEs* not only surpasses state-of-the-art baselines across various recognition tasks but also demonstrates zero-shot generalization to unseen, out-of-domain scenarios. We further demonstrate the transfer-learning strength of *ViTaPEs* in a robotic grasping task, where it outperforms state-of-the-art baselines in predicting grasp success. Project page: https://sites.google.com/view/vitapes

## 1 Introduction

Studies across species demonstrate that tactile perception is essential for the development and expression of intelligence, supporting perception, learning, and decision-making in living organisms (Banerjee et al., 2023; Diamond & Toso, 2023). For humans, touch is critical for tasks such as grasping, manipulation, material characterization, and detecting environmental changes (Lederman & Klatzky, 1987; Klatzky & Lederman, 1992). It provides essential information about object properties like texture, compliance, and force, which is vital for fine motor skills and subtle interactions (Calandra et al., 2017). Tactile sensing can offer local descriptions of deformation at contact points, providing information that other modalities cannot efficiently capture.

---

*Corresponding author: www.lygerakis.com

When combined with vision, it enhances perception by offering fine-grained details like pressure distributions and surface compliance, complementing vision's global view of object shapes and spatial relationships (Dahiya et al., 2010; Calandra et al., 2017). Together, these modalities provide a thorough contextual understanding of the environment.

Recent works in visuotactile representation learning have shown the potential for joint models, improving performance in complex tasks that rely on both modalities (Chen et al., 2022; Yang et al., 2024; Fu et al., 2024; Lygerakis et al., 2024). Although a number of these methods can effectively learn shared representations, challenges remain, including aligning data across different sensory scales and handling domain-specific artifacts (e.g., texture, compliance, localization, scene context) and explicitly modeling positional encodings for multi-stage spatial alignment of touch with vision.

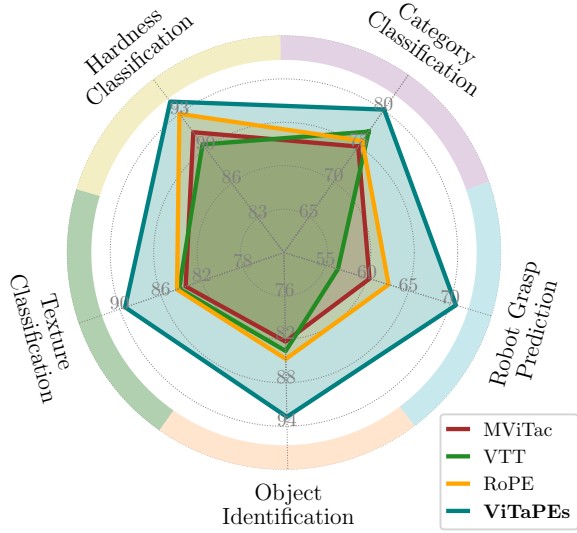

Figure 1: Task-accuracy radar comparing visuotactile models. *ViTaPEs* outperforms all others in robustness and cross-domain generalization.

Current research in this field often relies on large, pretrained visual or vision-language models (VLMs) (Yang et al., 2024; Fu et al., 2024), where the visual encoder is frozen and only the tactile encoder is trained to align with it. This can limit expressivity and assumes that visual representations are optimal for tactile alignment, hindering joint representation learning. A notable ViT-based exception is VTT (Chen et al., 2022), but it is trained solely on simulated force-torque feedback, far lower in resolution and complexity than data from high-resolution spatial sensors like DIGIT (Lambeta et al., 2020), GelSight (Yuan et al., 2017a), or uSkin (Tomo et al., 2018). Moreover, VTT depends on application-specific auxiliary losses, and its generalization to broader tasks remains untested.

Another key limitation is the narrow scope of most existing approaches. Current models are typically fine-tuned for specific downstream tasks, such as object manipulation, material classification, texture recognition, or cross-modal generation, reducing their ability to generalize (Dave et al., 2024; Jang et al., 2022; Fang et al., 2024; Yang et al., 2023). As a result, they often lack the versatility needed for broader applications. A more task-agnostic approach that performs well with little or no fine-tuning would significantly improve the practical utility of visuotactile representation learning.

We present a cross-modal method that integrates visual and tactile data using a transformer-based architecture enriched with multi-stage **Vi**suo-**Ta**ctile **P**ositional **E**ncoding**s**, namely *ViTaPEs*. In visuotactile settings, positional information matters at two complementary points in the computation graph. First, each modality carries its own spatial layout (e.g., local surface deformation patterns in touch or scene context in vision) that should remain distinguishable within the stream. Second, cross-modal interaction is realized when the two token sequences are processed jointly by self-attention; at this fusion stage it is beneficial to expose both modalities to a shared positional vocabu-

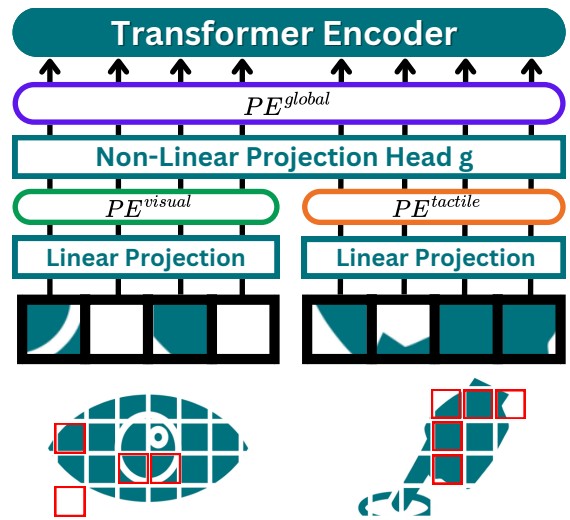

Figure 2: *ViTaPEs* Architecture: The visual and tactile inputs are projected into separate token spaces, followed by the addition of modality-specific (green and orange) and a shared (purple) global PEs for multi-modal fusion, injecting positional signals within each stream and on the joint sequence before attention.

lary so the model can learn correspondences during training, without assuming a geometrically calibrated alignment.

We operationalize this with two-stage positional injection. Modality-specific *local* encodings are added within each stream, then the visual and tactile token sequences are concatenated, and a single learned *global* positional encoding is added on the joint sequence immediately before attention. Since vanilla transformers (Vaswani et al., 2017) are permutation-equivariant without positional signals and large synchronized visuotactile datasets are scarce, this design supplies an inductive bias by injecting positional information (i) where token-wise feature extraction occurs and (ii) where cross-modal mixing occurs, guiding attention to discover useful cross-modal relationships without relying on prohibitively large training corpora.

Our model can be trained in both self-supervised and supervised regimes, facilitating task-agnostic embeddings while also optimizing for specific downstream tasks. Evaluations on out-of-distribution objects in real-world scenarios substantiate the efficacy of *ViTaPEs* in grasp success prediction, object recognition, material characterization, texture analysis, and hardness assessment. We contribute with:

- **Multi-Stage Positional Encodings:** Our *ViTaPEs* model employs a multi-stage positional design that encodes spatial structure within each modality and adds shared positional signals on the joint sequence before attention, supporting multi-stage positional reasoning in visuotactile fusion.

- **Zero-Shot Generalization and Transferable Representations:** We demonstrate the out-of-distribution generalization capacity of *ViTaPEs*, trained with self-supervision, highlighting its robustness across diverse tasks and environments. We further show that *ViTaPEs* outperforms baseline methods on a real-world robotic grasping dataset, leveraging its transfer learning capabilities to adapt effectively to a smaller dataset of 10K samples.

## 2 Related Work

**Visuotactile Representation Learning.** Visuotactile representation learning combines vision and touch to enhance perception in a wide range of tasks, including manipulation, material recognition, and texture analysis. Recent approaches have leveraged deep learning to jointly model visual and tactile data. Yuan et al. (2017b) introduced a shared latent space for the two modalities using GelSight sensors (Yuan et al., 2017a) for fabric classification. Building on this, Yang et al. (2022) and Kerr et al. (2023b) employed contrastive learning techniques to improve tactile representation learning with GelSight (Yuan et al., 2017a) and DIGIT (Lambeta et al., 2020) sensors, respectively. Li et al. (2019) addressed the scale gap between visual and tactile signals using conditional adversarial networks to synthesize tactile data from visual inputs. Luo et al. (2018) improved cloth texture recognition by focusing on shared features across modalities, while the Visuo-Tactile Transformer (VTT) (Chen et al., 2022) utilized spatial attention to effectively merge visual and tactile data. More recently, MViTac (Dave et al., 2024) demonstrated the effectiveness of multimodal contrastive training, learning both intra- and inter-modal representations to improve material classification and grasp prediction.

**Transformer-Based Multimodal Fusion.** Transformer-based architectures excel at modeling complex cross-modal relationships but often rely heavily on pre-trained large language models (LLMs) or vision-language models (VLMs), which limits their adaptability to visuotactile domains. Unitouch (Yang et al., 2024) aligns tactile data with embeddings from pre-trained VLMs, achieving multimodal alignment between language, vision, and touch. However, this comes under the assumption that the visual latent space is optimal, thereby overlooking tactile-specific richness. Similarly, Fu et al. (2024) leverages pre-trained LLMs and VLMs to align touch, vision, and language. This approach also forces tactile data to conform to representations optimized for other modalities, potentially constraining the expressivity and adaptability of the tactile features.

**Positional Encodings in Transformers.** Transformers rely on positional encodings to incorporate structural information, as they lack inherent inductive biases for sequential or spatial data. Absolute PEs, such as sinusoidal functions or learned embeddings, enable generalization to unseen sequences but fail to capture relational dependencies (Vaswani et al., 2017). Relative PEs address this limitation by modeling

relationships between elements based on their distances, improving relational reasoning tasks (Shaw et al., 2018). However, relative PEs are limited by their inability to generalize to arbitrary-length inputs and their increased computational complexity due to explicit pairwise distance calculations, making them less efficient for long or high-dimensional data. Rotary PEs (RoPE) (Su et al., 2024; Heo et al., 2025) address these limitations by encoding relative positions through rotating query and key vectors, offering a more efficient solution that scales effectively with sequence length. However, RoPE does not explicitly capture multi-stage spatial relationships or the complex positional dynamics needed for both effective multi-modal integration and detailed within-modality spatial nuances, limiting its applicability in tasks requiring comprehensive spatial understanding.

## 3  *ViTaPEs*: Visuotactile Positional Encodings

To address the limitations of existing approaches in visuotactile joint modeling, we propose **_ViTaPEs_** , a unified architecture for integrating visual and vision-based tactile data based on a vision transformer (ViT) architecture (Dosovitskiy et al., 2021). *ViTaPEs* incorporates multi-stage PEs to effectively capture both intra-modal and inter-modal relationships. Specifically, our multi-stage design consists of unimodal PEs (Section 3.1) that operate on individual modalities and a global PE (Section 3.2) shared across the concatenated visuotactile token sequence. By leveraging attention mechanisms (Vaswani et al., 2017), *ViTaPEs* models complex multimodal interactions, enabling robust joint representation learning to improve performance in tasks requiring integrated visual and tactile perception.

At the core of ViTaPEs is the ability to process visual and tactile data within a single transformer encoder. Each modality's input is patchified into tokens that carry its own spatial layout. However, unlike CNNs, transformers (Vaswani et al., 2017) do not exploit this structure by default. After patchification and projection, token embeddings encode only local content, not their position. To restore this information, we add modality-specific *local* positional encodings that preserve within-stream geometry. Second, the two streams are processed in a common fusion stage so that attention can model correspondences between visual and tactile tokens. We operationalize this by introducing a *global* cross-modal positional encoding that places tokens from both streams into a common reference before any attention layers mix them. Local PEs respect within-modality geometry, and the shared global PE supplies a shared positional vocabulary at the fusion stage, without assuming a geometrically calibrated coordinate system, thereby promoting stable alignment.

The visual input is represented as $\mathbf{V} \in \mathbb{R}^{N_{\text{visual}} \times P}$, where $N_{\text{visual}}$ denotes the number of visual patches, and $P$ is the dimensionality of each flattened patch. Similarly, the tactile input is represented as $\mathbf{T} \in \mathbb{R}^{N_{\text{tactile}} \times B}$, where $N_{\text{tactile}}$ denotes the number of tactile patches, and $B$ is the dimensionality of each flattened patch. These patches are mapped into an embedding dimension $D$ via learnable linear transformations to form tokens:

$$\mathbf{X}^{\text{visual}}(V) = \mathbf{V}\,\mathbf{W}^{\text{visual}}, \quad \mathbf{X}^{\text{tactile}}(T) = \mathbf{T}\,\mathbf{W}^{\text{tactile}} \tag{1}$$

where $\mathbf{X}^{\text{visual}} \in \mathbb{R}^{N_{\text{visual}} \times D}$ and $\mathbf{X}^{\text{tactile}} \in \mathbb{R}^{N_{\text{tactile}} \times D}$ are the token embeddings for the visual and tactile modalities, respectively. Here, $\mathbf{W}^{\text{visual}} \in \mathbb{R}^{P \times D}$ and $\mathbf{W}^{\text{tactile}} \in \mathbb{R}^{B \times D}$ are learnable weight matrices. These token embeddings serve as the initial representations for each modality.

### 3.1  Uni-modal Position Encodings

Each modality carries distinct spatial and semantic characteristics. For instance, standard visual images typically capture global spatial descriptors aligned with a camera-based view, whereas tactile images may encode sensor-specific signals such as pressure or contact distribution across a specialized surface. To accommodate these differences, we assign a separate learnable modal positional encoding to each modality, thereby providing a dedicated spatial representation for each domain. Specifically, the visual modality employs $\mathbf{PE}^{\text{visual}} \in \mathbb{R}^{N_{\text{visual}} \times D}$ and the tactile modality employs $\mathbf{PE}^{\text{tactile}} \in \mathbb{R}^{N_{\text{tactile}} \times D}$. To preserve modality-specific structure, each position encoding is added directly to its corresponding token:

$$\mathbf{X}^{\text{visual}}_{\text{modal}}(V) = \mathbf{X}^{\text{visual}}(V) + \mathbf{PE}^{\text{visual}}, \quad \mathbf{X}^{\text{tactile}}_{\text{modal}}(T) = \mathbf{X}^{\text{tactile}}(T) + \mathbf{PE}^{\text{tactile}}. \tag{2}$$

These modality-specific PEs enable the transformer encoder to capture the unique spatial priors inherent to each sensor, before any cross-modal mixing occurs.

## 3.2 Global Position Encoding

A key part of our design is a global positional encoding added to the joint token sequence immediately before self-attention, providing a shared positional vocabulary at the stage where cross-modal interaction is realized. Even though each modality has its own distinct layout, cross-modal tasks benefit from exposing attention to positional signals on the concatenated sequence when relating tokens across modalities.

We optimize a global positional encoding $\mathbf{PE}^{\text{global}} \in \mathbb{R}^{(C+N) \times D}$, where $N = N_{\text{visual}} + N_{\text{tactile}}$ is the total number of patch tokens across modalities, and $C$ is 1 if a classification token is included (otherwise 0).

**Token-axis concatenation and token-wise stem.** We first form the joint patch-token sequence by *token-axis* (row-wise) concatenation:

$$\mathbf{X}_{\text{concat}}(V, T) = \left[ \mathbf{X}_{\text{modal}}^{\text{visual}}(V) \, ; \, \mathbf{X}_{\text{modal}}^{\text{tactile}}(T) \right] \in \mathbb{R}^{N \times D}. \tag{3}$$

If a CLS token is used, we prepend it to obtain $\tilde{\mathbf{X}}_{\text{concat}}(V, T) \in \mathbb{R}^{(C+N) \times D}$ (otherwise $\tilde{\mathbf{X}}_{\text{concat}} = \mathbf{X}_{\text{concat}}$).

We denote by $g : \mathbb{R}^D \to \mathbb{R}^D$ a two-layer MLP applied *token-wise* (row-wise) with shared weights, and we apply it independently to each token:

$$\tilde{\mathbf{X}}_{\text{projected}}(V, T) = g\big(\tilde{\mathbf{X}}_{\text{concat}}(V, T)\big) \in \mathbb{R}^{(C+N) \times D}. \tag{4}$$

**Global positional encoding at the fusion stage.** We then add $\mathbf{PE}^{\text{global}}$ to obtain the sequence fed to the transformer:

$$\mathbf{X}_{\text{global}}(V, T) = \tilde{\mathbf{X}}_{\text{projected}}(V, T) + \mathbf{PE}^{\text{global}}\big[1 : (C + N), : \big]. \tag{5}$$

While the local and global positional encodings could theoretically be algebraically collapsed if the network were purely linear, their separation across the non-linear projection head $g$ is structurally intentional. Injecting local PEs before $g$ allows the optimizer to decouple the learning of non-linear geometric spatial warping (handled by $g$) from the addition of shared positional signals on the joint sequence immediately before attention (handled by $PE^{global}$). This separation creates two injection points that affect different parts of the computation graph: local PEs condition token-wise feature extraction inside $g$, while the global PE supplies positional signals on the joint sequence immediately before attention, where cross-modal mixing is realized.

## 3.3 Transformer Operations and Cross-Attention for Multi-Modal Integration

Once the positionally-encoded tokens $\mathbf{X}_{\text{global}}$ are obtained, they are fed into a single transformer encoder. The transformer architecture comprises standard layers, including multi-head self-attention and feed-forward networks, adapted to process the combined visual–tactile token sequence.

Given the fused embeddings $\mathbf{X}_{\text{global}}$, the self-attention mechanism can capture both intra-modal structure and cross-modal dependencies, where visual information provides context for tactile details and vice versa. The self-attention operation is defined as:

$$\text{Attention}(Q, K, V) = \text{Softmax}\Big(\frac{QK^T}{\sqrt{d_k}}\Big) V, \tag{6}$$

where $Q$, $K$, and $V$ are the query, key, and value matrices derived from the positionally-encoded tokens, and $d_k$ is the dimensionality of the key vectors for each head in the multi-head attention mechanism, i.e., $d_k = D/h$, where $h$ is the number of attention heads.

When the above attention mechanism is applied to the concatenation of visual and tactile tokens, it yields both self-attention within each modality and cross-attention across modalities (Chen et al., 2022). To make this explicit, we partition the token sequence into visual and tactile parts and write, for a given head,

$$Q = \begin{bmatrix} Q_i \\ Q_t \end{bmatrix}, \quad K = \begin{bmatrix} K_i \\ K_t \end{bmatrix}, \quad V = \begin{bmatrix} V_i \\ V_t \end{bmatrix},$$

Table 1: In-domain top-1 accuracy (%) evaluations across various downstream tasks, separated by models trained with and without SSL. We trained the models on the TAG dataset (Yang et al., 2022) for the category, hardness, and texture tasks, and on Object Folder Real (Gao et al., 2023) for "OF-Real" identification; "YCB" column reports performance on YCB-Slide. See Appendix H for details of baseline methods. We omit supervised results on YCB-Slide in Table 1 because the task saturates under supervision, precluding meaningful comparative analysis. Numbers are the mean over 5 seeds.

| Methods | Model Backbone | Material Property Recognition | | | Object Identification | | SSL | # params |
| | | Category | Hardness | Texture | OF-Real | YCB | | |
|---|---|---|---|---|---|---|---|---|
| Vanilla CNN | ResNet | 46.9 | 72.3 | 76.3 | 89.8 | − | ✗ | 22.3M |
| VTT (Chen et al., 2022) | ViT | 77.0 | 90.6 | 84.7 | 83.6 | - | ✗ | 12.0M |
| RoPE (Heo et al., 2025) | ViT | 75.7 | 93.6 | 84.9 | 84.7 | - | ✗ | 12.0M |
| *ViTaPEs (Ours)* | ViT | **80.1** | **94.8** | **89.7** | **92.7** | - | ✗ | 12.7M |
| TAG (Yang et al., 2022) | ResNet | 54.7 | 77.3 | 79.4 | 81.2 | 79.3 | ✓ | 22.3M |
| SSVTP (Kerr et al., 2023a) | ResNet | 70.1 | 88.6 | 83.6 | 53.8 | 76.7 | ✓ | 22.4M |
| MViTac (Dave et al., 2024) | ResNet | 74.9 | 91.8 | 84.1 | 82.3 | 91.5 | ✓ | 22.4M |
| VTT (Chen et al., 2022) | ViT | 72.4 | 88.2 | 83.3 | 76.8 | 85.5 | ✓ | 29.7M |
| RoPE (Heo et al., 2025) | ViT | 73.0 | 89.5 | 84.0 | 77.5 | 81.9 | ✓ | 29.7M |
| *ViTaPEs (Ours)* | ViT | **75.9** | **92.2** | **87.2** | **85.2** | **96.9** | ✓ | 30.6M |

where $(Q_i, K_i, V_i)$ are derived from visual tokens and $(Q_t, K_t, V_t)$ from tactile tokens. For the $j$-th head of the $n$-th attention layer, the attention output can be written as:

$$A_j^n = \text{Softmax}\Big(\frac{1}{\sqrt{d_k}} \begin{bmatrix} Q_i \\ Q_t \end{bmatrix} \begin{bmatrix} K_i \\ K_t \end{bmatrix}^T\Big) \begin{bmatrix} V_i \\ V_t \end{bmatrix} = \begin{bmatrix} A_{ii} & A_{it} \\ A_{ti} & A_{tt} \end{bmatrix} \begin{bmatrix} V_i \\ V_t \end{bmatrix} = \begin{bmatrix} A_{ii}V_i + A_{it}V_t \\ A_{ti}V_i + A_{tt}V_t \end{bmatrix}, \tag{7}$$

where the block matrices $A_{ii}, A_{it}, A_{ti}, A_{tt}$ denote the corresponding sub-blocks of the (post-softmax) attention matrix. Importantly, the softmax normalization is taken over the full joint token sequence, so these blocks are coupled through the shared denominator; the decomposition is for interpretability rather than a separability claim. The cross-modal interactions are captured by $A_{it}V_t$ (visual queries attending to tactile values) and $A_{ti}V_i$ (tactile queries attending to visual values).

The final output of the transformer encoder represents a comprehensive visuotactile feature map, preserving intra-modal relationships while capturing inter-modal dependencies. This feature map can be pooled or processed via a classification token for downstream tasks.

# 4 Experiments

We evaluate *ViTaPEs* on three types of tasks: material recognition, cross-sensor generalization, and robotic grasp prediction. Training is performed on TAG (Yang et al., 2022) and OF-Real (Gao et al., 2023), and we additionally test on YCB-Slide (Suresh et al., 2022) and unseen out-of-domain splits. For manipulation, we use the grasp prediction dataset from Calandra et al. (2018). Performance is measured by top-1 accuracy for classification. In the linear-probe setting, the encoder is frozen and only a linear classifier is trained, while in the zero-shot setting, we train on a source dataset and evaluate directly on a disjoint target without fine-tuning.

Unless otherwise noted, results are averaged over multiple random seeds. Implementation details, hyperparameters, and augmentations are provided in Appendix B, and we report training cost and inference throughput in Appendix D.

## 4.1 Material Property Recognition

We first assess the ability of *ViTaPEs* to capture material-specific features on the Touch-and-Go (TAG) dataset (Yang et al., 2022). Alongside standard supervised learning, we employ a masked autoencoder (MAE) (He et al., 2022) for self-supervised pre-training (SSL) of the ViT-based models, aiming to acquire

robust and task-agnostic representations. To ensure fair comparisons and match the model capacity of other self-supervised baselines, we employ our *Balanced* (see Appendix E for the model sizes) ViT encoder for the MAE. Further details on this architecture and our training choices are provided in Appendix B. The TAG dataset comprises 20 diverse objects, each providing tactile feedback indicative of distinct material properties (see Appendix G.1 for dataset details). Following Yang et al. (2022), we evaluate on three tasks: *Category* (classifying samples into 20 object types), *Hardness*, and *Texture* (both binary classification tasks). These tasks highlight how local tactile signals and global visual cues collectively characterize material attributes.

**Results**: Table 1 shows that CNN-based methods (TAG, Vanilla-CNN) underperform, suggesting that ResNet backbones may struggle to capture the multi-faceted aspects of material recognition. In contrast, ViT-based models (VTT, RoPE, and *ViTaPEs*) leverage attention and PEs more effectively, boosting performance across all tasks. Notably, *ViTaPEs*, with its multi-stage PEs, consistently exceeds the accuracy of VTT and RoPE, demonstrating the benefit of leveraging (i) within-modality absolute layout cues (local PEs) and (ii) fusion-stage positional cues on the joint sequence (global PE). Under supervised training, *ViTaPEs* reaches 80.1% in Category, 94.8% in Hardness, and 89.7% in Texture, largely outperforming competing approaches.

When integrating self-supervision, all models experience performance changes, though the extent varies. It is important to note that for ViTaPEs, the supervised model is trained end-to-end with a non-linear MLP head, whereas the SSL performance reflects a frozen-encoder evaluated via a single linear classifier (linear probe). Because the SSL encoder is optimized for task-agnostic visuotactile structure rather than being explicitly fine-tuned for the TAG Category task, its in-domain accuracy is slightly lower than its fully supervised counterpart. However, this task-agnostic representation yields vastly superior out-of-distribution transfer, as demonstrated in subsequent sections. ResNet-based methods appear to benefit from SSL, however, their gains remain modest due to the limited capacity of CNN architectures in capturing complex cross-modal interactions. VTT and ROPE show a moderate decrease with SSL, falling behind ViTaPEs. Notably, ViTaPEs achieves the highest performance across all tasks in the self-supervised setting, reaching 75.9% in Category, 92.2% in Hardness, and 87.2% in Texture.

## 4.2 Object Identification

We evaluate *ViTaPEs* on object identification using two benchmarks: the Object Folder Real (OF-Real) dataset (Gao et al., 2023) and the YCB-Slide (YCB) dataset (Suresh et al., 2022). OF-Real features real-world household objects spanning materials such as wood, glass, plastic, and steel, with tactile readings collected via the GelSight robotic finger (Yuan et al., 2017a) aligned to synchronized video frames. YCB comprises 21 common objects captured with paired RGB and DIGIT tactile images under varying lighting and backgrounds, serving as a cross-sensor transfer test for generalization. Both benchmarks challenge models to integrate localized tactile signals with global visual features, making them ideal for assessing cross-modal alignment capabilities.

**Results:** As shown in Table 1, ResNet-based approaches fail to effectively capture the multi-modal complexities of object identification, compared to the ViT counterparts, achieving relatively lower accuracy on OF-Real. Transformer-based models such as VTT and RoPE improve on these baselines but still lag behind *ViTaPEs*, which achieves 92.7% top-1 accuracy in the supervised regime and 85.2% in the SSL setting. Importantly, on the YCB dataset, used here as a cross-sensor transfer benchmark, *ViTaPEs* attains 96.9% top-1 accuracy under SSL, outperforming the next-best method by over 5%, and demonstrating exceptional generalization across both datasets.

## 4.3 Zero-Shot Generalization

To evaluate out-of-domain generalization, we test models pre-trained with SSL on one dataset and evaluate their performance on a different one without additional fine-tuning.

We conduct two types of evaluations: ***linear probing*** and ***zero-shot***. In linear probing, a linear classifier is trained on top of the frozen encoders to assess how well the learned representations transfer across datasets. For zero-shot evaluation, cosine similarity is computed directly from the frozen encoders without additional training. Table 2 summarizes the results for both setups across the TAG and OF-Real datasets. All ViT-based

Table 2: Cross-dataset transfer accuracies (%) under *linear probing* and *zero-shot* evaluation. Columns correspond to the two transfer directions: OF-Real→TAG (reported under "TAG") and TAG→OF-Real (reported under "OF-Real"). The *N/A* entries for UniTouch reflect that we evaluate the released checkpoint but do not retrain this large-scale multi-dataset pretrained model under the controlled single-source OF-Real→TAG setting.

| Methods | Linear Probe | | Zero-Shot | |
|---|---|---|---|---|
| | TAG | OF-Real | TAG | OF-Real |
| MViTac (Dave et al., 2024) | 48.3 | 50.2 | 39.5 | 41.8 |
| UniTouch (Yang et al., 2024) | *N/A* | 61.2 | *N/A* | 33.2 |
| SigLIP2 (Tschannen et al., 2025) | 51.4 | 56.0 | 47.6 | 36.1 |
| VTT (Chen et al., 2022) | 49.7 | 55.0 | 41.8 | 45.4 |
| RoPE (Heo et al., 2025) | 47.1 | 52.4 | 40.0 | 44.3 |
| ***ViTaPEs*** | **53.1** | **68.1** | **53.8** | **65.2** |

Table 3: Performance on the Grasp dataset for success prediction. We report accuracy (%) for *SSL* (fine-tune and evaluate on Grasp), and transfer via *Linear* (linear probing) and *Zero* (zero-shot) after pretraining on TAG. For large pretrained VLM-style baselines, we report only Linear/Zero results.

| Methods | SSL | Linear | Zero |
|---|---|---|---|
| SSVTP (Kerr et al., 2023a) | 58.2 | 59.1 | 53.6 |
| TAG (Yang et al., 2022) | 56.3 | 57.8 | 50.9 |
| MViTac (Dave et al., 2024) | 60.3 | 60.7 | 56.0 |
| UniTouch (Yang et al., 2024) | *N/A* | 56.0 | 56.5 |
| SigLIP2 (Tschannen et al., 2025) | *N/A* | 56.4 | 53.1 |
| VTT (Chen et al., 2022) | 56.5 | 62.9 | 55.2 |
| RoPE (Heo et al., 2025) | 62.6 | 64.5 | 57.4 |
| ***ViTaPEs*** | **70.7** | **69.3** | **60.4** |

baselines (VTT, RoPE, *ViTaPEs*, and SigLIP2) are pre-trained on the source dataset using the same SSL recipe; SigLIP2 differs only in that its visual backbone is initialized from a VLM-pretrained checkpoint and is fine-tuned with a smaller lr ($10^{-5}$). UniTouch is evaluated via its released checkpoint and is not retrained under the controlled single-source settings (Appendix H).

**Results:** *ViTaPEs* achieves the strongest transfer in both directions and under both evaluation modes. On OF-Real (pre-trained on TAG), *ViTaPEs* reaches 68.1% linear and 65.2% zero-shot, outperforming UniTouch (61.2% / 33.2%) and other transformer baselines (VTT, RoPE). On TAG (pre-trained on OF-Real), *ViTaPEs* also leads (53.1% / 53.8%), improving over SigLIP2 (51.4% / 47.6%). We include UniTouch as a strong, large-scale pretrained baseline using the released checkpoint; however, retraining it under the controlled single-source setting (OF-Real→TAG) is not applicable, and we therefore report *N/A* for those entries.

These transfers involve a pronounced domain shift: TAG and OF-Real employ tactile sensors with different gels and capture conditions, yielding substantially different tactile image statistics. For qualitative intuition, Appendix Figure 4 shows paired visual–tactile frames across datasets, highlighting the appearance gap induced by sensor morphology and illumination. The strong zero-shot performance of *ViTaPEs* indicates that its learned visuotactile representations remain stable under this sensor-induced shift, supporting deployment in settings with heterogeneous tactile hardware.

### 4.4 Robot Grasping Prediction

We evaluate our approach on the Grasp dataset (Calandra et al., 2018) too, a benchmark for predicting grasp success or failure using tactile data from a parallel-jaw gripper and RGB images. For details on the dataset and its preprocessing configurations, we refer the readers to Appendix G.4.

To assess the learned visuotactile representations, we report three evaluation schemes: *SSL*, *Linear* (linear probing), and *Zero* (zero-shot). In all settings, models are initialized from pre-training on the out-of-distribution TAG dataset (Yang et al., 2022). In the *SSL* setup, we fine-tune the full model on Grasp and evaluate on Grasp. In the *Linear* and *Zero* setups, we freeze the encoder and follow the same linear-probe and cosine-similarity evaluation protocol as in Section 4.3. Additional details are provided in Appendix F.

**Results:** Table 3 shows that *ViTaPEs* achieves the best performance across all reported settings. It improves over ResNet-based baselines (SSVTP, TAG, MViTac) and transformer baselines (VTT, RoPE), reaching 70.7% with SSL fine-tuning, 69.3% with linear probing, and 60.4% in the zero-shot transfer setting. Moreover, *ViTaPEs* also surpasses VLM-initialized baselines (UniTouch, SigLIP2) under linear and zero-shot evaluation. These gains are particularly pronounced in this low-data regime (approximately 10K Grasp samples), where strong inductive biases and transferable visuotactile features are crucial for reliable generalization.

## 5 Ablation Studies and Discussion

The *ViTaPEs* approach sets a new state-of-the-art across a wide range of tasks: in Section 4.1 it excels in all baselines on material category, hardness, and texture; in Section 4.2, it achieves top-1 accuracy on object identification on both OF-Real and YCB; and in Sections 4.3 and 4.4 it demonstrates exceptional transfer via linear probing and zero-shot evaluation, surpassing pre-trained VLM baselines such as UniTouch (Yang et al., 2024) and SigLIP2 (Tschannen et al., 2025). These results confirm that our novel multi-stage PEs enable robust visuotactile feature fusion and generalization.

**Learnable or Sinusoidal Positional Encodings?** Table 4 examines how different choices for PEs, learnable vs. sinusoidal, affect performance on the *Category* task. The three left columns compare replacing the learnable *local* (modality-specific) and/or *global* positional encodings with fixed sinusoidal encodings.

We observe that relying on sinusoidal encodings for either modal or global PEs yields lower accuracy (76.2–76.5%) compared to fully learnable PEs. This confirms that learnable embeddings adapt more effectively to the peculiarities of visual-tactile data. Although sinusoidal encodings can provide a reasonable inductive bias, the dynamic nature of tactile signals, along with variations in camera perspectives, appears to benefit from the flexibility of learnable positional parameters.

Table 4: Ablation of positional encodings (PEs) and modality settings on the TAG (Yang et al., 2022) category task. The first three groups of columns toggle learnable vs. sinusoidal PEs (non-learnable = sinusoidal), whether each modality uses any PE, and whether vision or touch is active. The rightmost column reports the full *ViTaPEs* configuration with all learnable PEs and both modalities enabled, corresponding to the 80.1% result in Table 1. Numbers are the mean over 5 seeds.

|  | Learnable PE | | | PE Use | | Modality Use | | **ViTaPEs** |
|---|---|---|---|---|---|---|---|---|
| Visual | ✗ | ✓ | ✗ | ✗ | ✓ | ✓ | ✗ | |
| Tactile | ✗ | ✓ | ✗ | ✗ | ✓ | ✗ | ✓ | |
| Global | ✓ | ✗ | ✗ | ✓ | ✗ | | | |
| Category | 76.5 | 76.4 | 76.2 | 76.9 | 77.2 | 70.5 | 63.8 | **80.1** |

**Are Both Modal and Global PEs Effective?** The ablation in Table 4 reveals that omitting either modality-specific or global PEs drops accuracy to around 76.9–77.2% (two central columns). This indicates that each component encodes distinct spatial cues critical for aligning visual and tactile features. By contrast, employing both (rightmost column) raises accuracy to 80.1%, demonstrating the complementarity of encoding modality-specific structure (local PEs) alongside positional signals on the joint sequence at the fusion stage (global PE). In other words, our multi-stage design, where separate PEs are learned for the visual, tactile, and global contexts, fosters richer and more discriminative cross-modal representations.

Figure 3 underscores how each learned PE contributes differently. The *visual PE* (Fig. 3(left)) exhibits a grid-like pattern, capturing broad spatial dependencies aligned with typical camera-based inputs. Meanwhile,

the *tactile PE* (Fig. 3(middle)) shows higher-frequency fluctuations, reflecting its attention to localized texture variations and contact points vital for tactile sensing. Most notably, the *global PE* (Fig. 3(right)) presents a smooth overarching pattern, yet internal variations suggest it has also learned distinct nuances from each modality: the segments attending to vision (tokens 1- 196) appear more uniformly distributed, while those focusing on tactile signals exhibit finer oscillations. By exposing both modalities to a shared positional vocabulary at the cross-modal mixing stage, *ViTaPEs* effectively leverages large-scale visual context alongside the subtle deformations captured by tactile feedback. This complementary encoding ultimately drives the performance gains seen in Table 1, confirming that local (within-modality) cues and fusion-stage positional signals both play essential roles in robust visuotactile representation learning.

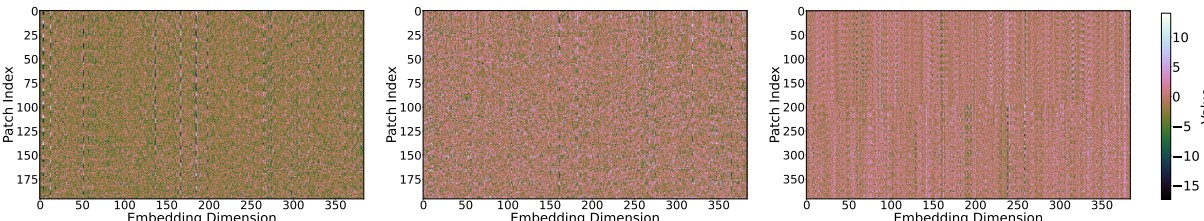

Figure 3: Learned PEs in *ViTaPEs* after training: visual, tactile, and global (left to right). Each PE exhibits a unique spatial structure reflecting modality-specific priors and representational needs.

**Role of the Projection Head and Injection Point.** To explicitly isolate the effect of our two-stage positional injection, we ablate the placement and non-linearity of the projection head $g$. As detailed in Table 5, we compare the full ViTaPEs model (where local PE is injected before a non-linear $g$) against three controlled variants: (i) adding the local PE *after* $g$ (PE-after-$g$), (ii) a parameter-matched control where the activation function is removed so $g$ is linear (Linearized-$g$), and (iii) removing the stem entirely (No-$g$). All variants are trained and evaluated under identical protocols and compute budgets.

Table 5: Ablation of the projection head ($g$) and the positional encoding injection point. Results show top-1 accuracy (%) for in-domain TAG Category on SSL, and out-of-domain transfer via linear probing and zero-shot evaluation on the OF-Real dataset. Numbers are the mean over 5 seeds.

| Variant | TAG Category | OF-Real Linear Probe | OF-Real **Zero-Shot** |
|---|---|---|---|
| ViTaPEs | **75.9** | **68.1** | **65.2** |
| PE-after-$g$ | 71.3 | 56.2 | 45.7 |
| Linearized-$g$ | 72.4 | 54.7 | 51.4 |
| No-$g$ | 70.9 | 52.9 | 48.1 |

Performance is strictly maximized when $g$ is non-linear and the local PE is injected *before* it. Crucially, the performance gap between ViTaPEs and the parameter-matched Linearized-$g$ control isolates the effect of the non-linear conditioning, proving that the gain is not simply due to added parameter capacity. This specific injection mechanism allows the network to learn position-conditioned feature extraction, which is critical for robust out-of-domain transfer under compound sensor shifts.

**Are Both Vision and Touch Important?** Another typical question that arises when developing visuotactile models is the benefit of using both modalities for solving tasks. As shown in the two right columns of Table 4, models not leveraging both vision and touch achieve lower accuracy compared to those using both modalities (80.1%). This demonstrates that integrating global visual context with localized tactile information enhances the model's discriminative capabilities, highlighting the complementary strengths of both vision and touch in visuotactile perception.

**Sensitivity to Data Variations** We assess robustness to missing tactile evidence on TAG by masking at evaluation a random fraction $p \in \{0, 20, 40, 60, 80, 100\}\%$ of tactile image patches, while keeping the vision

Table 6: TAG (Yang et al., 2022) category accuracy (%) under tactile-patch masking at evaluation (linear probe; encoders frozen after masked-autoencoding pretraining with 75% masking; 5 seeds per $p$). Higher is better.

|  | 0% | 20% | 40% | 60% | 80% | 100% |
|---|---|---|---|---|---|---|
| VTT | 72.4 | 72.2 | 72.0 | 71.8 | 69.8 | 66.7 |
| RoPE | 73.0 | 73.0 | 72.9 | 72.6 | 70.4 | 66.2 |
| *ViTaPEs* | **75.9** | **75.2** | **74.9** | **74.6** | **72.4** | **68.1** |

stream intact. The encoder is frozen after masked-autoencoding pretraining (75% masking), and a single linear classifier is trained on top of it; for each $p$ we report the mean over five seeds.

As shown in Table 6, accuracy degrades monotonically with increasing $p$ for all methods, yet *ViTaPEs* remains best at every level, with VTT and RoPE trailing across the entire range. *ViTaPEs* preserves near-optimal performance with up to 40% missing tactile input and retains a 1.4% margin over VTT and RoPE even when all tactile pixels are removed. This robustness mirrors the masking ratio used during pretraining and highlights the advantage of our multi-stage positional encoding, which encourages redundancy across visual and tactile channels.

**Scalability and Efficiency of *ViTaPEs***   *ViTaPEs* scales predictably with encoder capacity while remaining computationally efficient. Increasing the embedding dimension, depth, and attention heads from the *Minimal* (6.7M parameters) to the *Extended* (90.7M parameters) variant results in an 18% gain in top-1 accuracy on the TAG category task. The *Balanced* model (30.6M parameters), used in our main experiments, offers the best trade-off between performance and model size (see Appendix E). Training costs remain reasonable at 62.5 GPU-hours (h) on an NVIDIA A100 40 GB, comparable to other ViT baselines like VTT (58.9 h) and RoPE (66.9 h) (see Appendix D.2). At inference, *ViTaPEs* runs in 10 ms per visual–tactile pair on an NVIDIA RTX 4090, with complexity $O(N^2 D)$ (see Appendix D.1). These results show that *ViTaPEs* is both scalable and deployment-friendly.

**Limitations**   Despite the explicit architectural design and controlled ablations, the *ViTaPEs* architecture has several limitations. First, its reliance on camera-based tactile sensors that appear in the datasets, like GelSight (Yuan et al., 2017a) and DIGIT (Lambeta et al., 2020), introduces biases, potentially limiting generalizability to other platforms with different resolutions or properties. Second, our scaling study is confined to encoders of up to 90 M parameters due to a 62 GPU-hour per-run budget, which precludes assessment of typical larger ViT scales (e.g., Base, Large (Dosovitskiy et al., 2021)) or more efficient transformer variants that could further enhance performance or latency. Lastly, the limited availability of diverse, synchronized multimodal datasets restricts the exploration of larger and more generalizable models.

## 6   Conclusion

In this paper, we introduced *ViTaPEs*, a transformer-based architecture enriched with multi-stage visuotactile positional encodings with explicit injection points (before a token-wise nonlinearity and immediately before attention) that are validated via controlled ablations. Extensive experiments demonstrate that *ViTaPEs* sets new state-of-the-art accuracy on material-property recognition, object identification, and robot-grasp success prediction while achieving the first strong zero-shot transfer across disparate sensors and demonstrating robustness to sensor dropout. By effectively integrating visual and tactile signals through both modality-specific and global PEs, *ViTaPEs* consistently outperforms state-of-the-art baselines in accuracy, robustness, and cross-domain adaptability. These results highlight the potential of multi-stage PEs in enhancing cross-modal alignment and representation learning. Looking ahead, we plan to explore scaling *ViTaPEs* to larger transformer architectures to further boost performance and applicability in more complex scenarios, including closed-loop robotic manipulation.

**Author Contributions**

Fotios Lygerakis contributed to the conceptualization of the work, the design and implementation of the proposed method, the experimental setup, data processing, model training, evaluation, analysis of results, visualization, and writing of the manuscript. Ozan Özdenizci contributed to the formulation of the research direction, provided technical feedback, supported the interpretation and presentation of the results, and contributed to the refinement of the manuscript. Elmar Rückert supervised the research and reviewed the final manuscript.

**Acknowledgments**

This work was carried out within the projects KIRAMET (FO999899661), MUTAVIA (FO999922732) and NNATT (FO999907606), funded by the Austrian Research Promotion Agency (FFG) under the programme of the Federal Ministry for Climate Action, Environment, Energy, Mobility, Innovation and Technology.

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

# A   Implementation details and tensor shapes

This appendix summarizes tensor shapes and the fusion specification used in all experiments, using the same notation as Section 3.

**Patch tokens.**   Let $\mathbf{X}^{\text{visual}} \in \mathbb{R}^{N_{\text{visual}} \times D}$ and $\mathbf{X}^{\text{tactile}} \in \mathbb{R}^{N_{\text{tactile}} \times D}$ denote the visual and tactile patch tokens after patch embedding (Eq. 1), where $N_{\text{visual}}$ and $N_{\text{tactile}}$ are the number of patches per modality and $D$ is the token dimension.

**Local (modality-specific) positional encodings.**   We use learnable absolute positional encodings $\mathbf{PE}^{\text{visual}} \in \mathbb{R}^{N_{\text{visual}} \times D}$ and $\mathbf{PE}^{\text{tactile}} \in \mathbb{R}^{N_{\text{tactile}} \times D}$ and form the modality-conditioned tokens (Eq. 2):

$$\mathbf{X}^{\text{visual}}_{\text{modal}} = \mathbf{X}^{\text{visual}} + \mathbf{PE}^{\text{visual}}, \qquad \mathbf{X}^{\text{tactile}}_{\text{modal}} = \mathbf{X}^{\text{tactile}} + \mathbf{PE}^{\text{tactile}}.$$

**Token-axis concatenation and CLS.**   We concatenate along the token axis (Eq. 3):

$$\mathbf{X}_{\text{concat}} = \left[\mathbf{X}^{\text{visual}}_{\text{modal}}; \mathbf{X}^{\text{tactile}}_{\text{modal}}\right] \in \mathbb{R}^{(N_{\text{visual}} + N_{\text{tactile}}) \times D}.$$

If a CLS token is used, we prepend it to obtain $\tilde{\mathbf{X}}_{\text{concat}} \in \mathbb{R}^{(C+N) \times D}$ with $C = 1$ (otherwise $C = 0$ and $\tilde{\mathbf{X}}_{\text{concat}} = \mathbf{X}_{\text{concat}}$), where $N = N_{\text{visual}} + N_{\text{tactile}}$.

**Token-wise stem and global positional encoding.**   The stem $g : \mathbb{R}^D \to \mathbb{R}^D$ is applied *token-wise* (row-wise) with shared weights to form $\tilde{\mathbf{X}}_{\text{projected}} = g(\tilde{\mathbf{X}}_{\text{concat}}) \in \mathbb{R}^{(C+N) \times D}$ (Eq. 4). A single learned global positional encoding $\mathbf{PE}^{\text{global}} \in \mathbb{R}^{(C+N) \times D}$ is then added immediately before self-attention (Eq. 5):

$$\mathbf{X}_{\text{global}} = \tilde{\mathbf{X}}_{\text{projected}} + \mathbf{PE}^{\text{global}}.$$

# B   Architecture and Training Details

All baselines process visual and tactile inputs as $224 \times 224$ RGB images. For ViT-based architectures, these inputs are divided into $16 \times 16$ non-overlapping patches, resulting in 196 patches per modality. Each patch is linearly projected into an embedding space of dimension $D = 384$, with modality-specific tokens further enriched using positional encodings.

For multi-stage positional fusion, we employ a non-linear projection layer implemented as:

$$\text{Linear}(D, D_h) \to \text{LeakyReLU} \to \text{Linear}(D_h, D),$$

where $D_h = 768$ is the hidden dimension. The transformer encoders use standard ViT components, including multi-head self-attention and feed-forward networks. The multi-head self-attention mechanism comprises $h$ attention heads, where each head operates on a subspace of dimensionality $d_k = D/h$, facilitating complex cross-modal interactions. The feed-forward networks are implemented as two-layer perceptrons with GeLU activations, enabling non-linear transformations of the attention outputs. Layer normalization and residual connections are applied throughout to stabilize training.

The architecture scales with $L$ transformer layers, configured as 6 layers for supervised tasks and 12 layers for self-supervised pre-training. We use the same number of heads $h$ as layers $L$ in each setting. We train the models in both supervised and self-supervised (SSL) paradigms using a learning rate of $1 \times 10^{-4}$, weight decay of 0.1, and a cosine warmup scheduler. For supervised classification tasks (e.g., *Category*, *Hardness*, *Texture*), we employ a batch size of 64, and random augmentation (Cubuk et al., 2020) for data augmentation. For SSL training with MAE, target an effective batch size of 1024 via gradient accumulation, and apply random resized cropping as the augmentation strategy. A masking ratio of 75% is used to promote robust representation learning. Crucially, these augmentations are applied independently to the visual and tactile streams. This deliberately breaks pixel-level spatial alignment during training, forcing the model to rely on global, shift-consistent structural correspondences rather than fixed geometric calibration.

## C  Data Augmentation Strategies

In addition to investigating positional encodings and modality usage, we also examine how different data augmentation strategies impact performance. As shown in Table 7, *Random Augmentation* leads to stronger results in supervised classification, reaching a top-1 accuracy of 80.1% on the *Category* task. Conversely, when training with masked autoencoders in a self-supervised setting, *Random Resized Crop* is more effective, achieving 75.9% compared to 61.3% under Random Augmentation. These findings highlight the need to tailor augmentation strategies to the training paradigm, as the requirements of supervised learning can differ considerably from those of self-supervised objectives.

Table 7: Ablation of augmentation strategies under self-supervised (SSL) pretraining and supervised linear probing on the TAG Category task.

| Augmentation Strategy | SSL | Category |
|---|---|---|
| Random Resized Crop | ✓ | 75.9 |
|  | ✗ | 76.9 |
| Random Augmentation | ✓ | 61.3 |
|  | ✗ | 80.1 |

## D  Training & Inference Details

### D.1  Inference Throughput and Complexity

*ViTaPEs* processes a fused visual-tactile input in 10 ms on an RTX 4090, equivalent to 100 pairs/s throughput. The transformer's runtime complexity is $O((C + N)^2 D)$, where $C \in \{0, 1\}$ indicates an optional CLS token, $N = N_v + N_t$ is the number of patch tokens, and $D$ is the embedding dimension.

### D.2  Training Cost Comparison

Table 8 reports the GPU-hour budget required to pre-train each model under identical hardware and data settings. ResNet-based baselines (SSVTP, MViTac, TAG) cluster around 40 GPU-h, while ViT-based methods are costlier: VTT needs 58.9 h, RoPE 66.9 h, and *ViTaPEs* 62.5 h. Thus, *ViTaPEs* sits between the two transformer baselines, slightly slower than VTT yet 7 % faster than RoPE, while delivering the strongest performance (see Table 1 in the main text).

Table 8: GPU-hour budget for pre-training under identical settings.

| Model | Backbone | GPU-hours |
|---|---|---|
| SSVTP | ResNet | 38.9 |
| MViTac | ResNet | 40.3 |
| TAG | ResNet | 39.2 |
| VTT | ViT | 58.9 |
| RoPE | ViT | 66.9 |
| ***ViTaPEs*** | ViT | **62.5** |

## E  Model Scaling Analysis

We vary the ViT backbone along three axes—embedding dimension ($D$), depth ($L$), and number of heads ($h$)—holding all other hyperparameters and pretraining protocol fixed.

Performance grows monotonically with parameter count, with diminishing returns beyond the *Balanced* model, the configuration used in our main experiments. Crucially, no architecture-specific tuning is required: the

Table 9: Effect of encoder scale on TAG category accuracy.

| Variant | Batch | $D$ | $L$ | $h$ | Parameters (M) | Accuracy (%) |
|---------|-------|-----|-----|-----|----------------|--------------|
| Minimal | 64 | 384 | 3 | 3 | 6.7 | 60.9 |
| Moderate | 64 | 384 | 6 | 6 | 12.9 | 67.1 |
| **Balanced** | 64 | 384 | 12 | 12 | 30.6 | 75.9 |
| Extended | 64 | 768 | 12 | 12 | 90.7 | 79.0 |

same multi-stage positional encoding delivers robust gains across all scales, underscoring *ViTaPEs'* versatility for both resource-constrained and high-capacity deployments.

## F  Transfer Learning Evaluation Protocol

We evaluate the transfer learning capabilities of *ViTaPEs* and baseline models on the robotic grasping task using three evaluation schemes: *SSL*, *linear probing*, and *zero-shot*. In the *SSL* setup, models are initialized with the pre-trained encoders from Table 1, trained on the Touch and Go (TAG) dataset (Yang et al., 2022), and then fine-tuned on the Grasp dataset (Calandra et al., 2018) using the standard SSL method as described in Section B with a learning rate of 0.001. This process enables the models to adapt their visuotactile representations to grasping-specific features while retaining knowledge from pretraining. For *linear probing*, we freeze the same pre-trained encoders and train a simple linear classifier on top using labeled data from the Grasp dataset, evaluating how well the learned representations transfer with minimal adaptation. In the *zero-shot* setting, we assess the frozen encoders without any additional training, using cosine similarity between test and reference embeddings to predict grasp success. These evaluation strategies provide a comprehensive assessment of *ViTaPEs'* adaptability to new tasks and its effectiveness in real-world transfer learning scenarios.

## G  Datasets

### G.1  Touch-and-Go Dataset

The Touch-and-Go (TAG) dataset (Yang et al., 2022) features paired visual and tactile data captured in naturalistic settings, with tactile sensors interacting with various objects while simultaneously recording egocentric video. This dataset encompasses approximately 13,900 tactile interactions involving around 4,000 unique objects across 20 material categories, providing a diverse range of real-world scenarios and tactile features essential for distinguishing material properties.

We evaluate the performance of *ViTaPEs* on the TAG dataset to address the task of material property identification. Specifically, we consider three downstream tasks: (1) categorization of materials into 20 distinct classes, (2) binary classification of hard versus soft surfaces, and (3) binary classification of smooth versus textured surfaces. For consistency, we adhere to the dataset splits prescribed by the authors of (Yang et al., 2022), ensuring that our evaluations are directly comparable to prior work and their baselines.

### G.2  Object Folder Dataset

The ObjectFolder Real (OF-Real) dataset (Gao et al., 2023) provides comprehensive multisensory data for 100 common household items. Each object is documented through high-quality 3D meshes, HD rotation videos, and multiple tactile recordings from a GelSight sensor (Yuan et al., 2017a). The tactile recordings detail gel deformations upon contact, complemented by in-hand and third-view camera angles, enabling a comprehensive analysis of tactile and visual interplay.

For this work, we selected a balanced subset of 50 objects from the full dataset. The selection was performed to ensure representative coverage across different material types (e.g., ceramic, wood, glass, metal) and object

categories (e.g., bowls, plates, utensils). This subset preserves the overall balance of material and object diversity present in the full dataset.

The selected objects are as follows:

- **Bowls, Plates, and Utensils:** Soup Spoon, Bowl, Salad Plate, Dinner Plate, Blue Bowl, Decorative Plate, Mixing Bowl, Serving Bowl, Soup Bowl.

- **Kitchen Tools:** Cutting Boards (Large, Middle, Small), Mixing Bowls (Large, Middle, Small), Fruit Bowl, Fork (Small, Large), Spoon (Small, Large), Knife (Large, Middle, Small).

- **Household Items:** Wine Glass, Drinking Cup, Beer Mug, Soup Ladle, Serving Spoon, Salad Fork, Mixing Spoon, Shovel Toy, Handle Spoon, Round Plate.

- **Miscellaneous:** Wrenches (Small, Middle, Large), Pestle, Mortar, Flowerpot (Large, Small), Sculpture, Display Stand.

The dataset was split datapoint-wise into 80% training and 20% testing data. Both vision and tactile modalities were utilized in this study to exploit the multisensory nature of the dataset.

### G.3   YCB-Slide Dataset

The YCB-Slide dataset (Suresh et al., 2022) provides aligned RGB–tactile sliding interactions for 10 standard YCB objects (e.g., sugar box, tomato soup can, mustard bottle, bleach cleanser, mug, power drill, scissors, adjustable wrench, hammer, baseball). Data were captured by moving each object over a fixed DIGIT sensor mount, yielding over 180 000 frames pairing a 224×224 RGB crop with a 64×64 tactile depth image under varied lighting and background conditions.

### G.4   Grasp Dataset

The Grasp dataset Calandra et al. (2018) provides paired tactile data and RGB images captured during grasping attempts. Tactile data is collected from two sensors attached to the left and right jaws of a parallel-jaw gripper, and each trial includes three frames: *before*, *during*, and *after* the grasp. The task is to predict whether the grasp will succeed or fail based on the *during* frame, as it provides the most relevant information about the outcome.

The dataset includes 106 objects, and for this work, we created a randomized datapoint-wise split of 80% for training and 20% for testing. To ensure data quality, we retained only demonstrations with sufficient grasping attempts, balancing successful and failed grasps. For fine-tuning (SSL) the models in Section 4.4, the tactile images from the two sensors were concatenated across the channel dimension, resulting in a 6-channel tactile input. For the linear and zero-shot probing, we used only one of the two tactile images as input (3-channel) to the encoder. The *during* frame (mid-grasp) is used for prediction, as it provides the most relevant information regarding the success of the attempt.

### G.5   Dataset Examples

Figure 4 showcases one RGB frame (left) and the corresponding tactile image (right) for each dataset used in our study; TAG, Object Folder Real (OF-Real), Grasping, and YCB-Slide. Note the markedly different gel textures, marker layouts, colour palettes, and contact footprints produced by the three generations of GelSight sensors (TAG, OF-Real, Grasping) versus the DIGIT sensor (YCB-Slide). These domain shifts motivate the cross-sensor experiments in Section 4.2.

## H   Baselines

To benchmark the performance of *ViTaPEs*, we compare it against several state-of-the-art baselines, spanning convolutional neural networks (CNNs), Vision Transformers (ViTs), and specialized visuotactile models. These

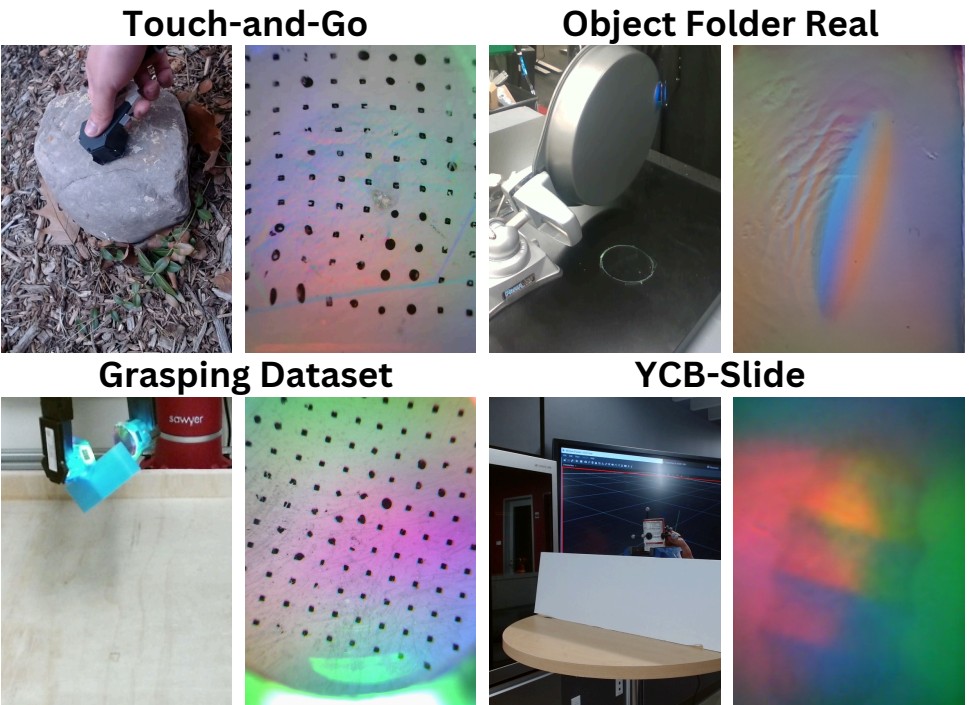

Figure 4: Paired visual (left) and tactile (right) samples across datasets, illustrating the heterogeneous visual appearance and tactile signal characteristics that *ViTaPEs* must handle.

baselines include ResNet-based architectures, self-supervised frameworks, and models designed specifically for tactile or visuotactile tasks. Below, we provide an overview of each baseline:

**Vanilla CNN**   We use a widely used CNN backbone with 18 layers and residual connections. It serves as a baseline for tasks involving visual or tactile data, with separate branches used for each modality when applied to visuotactile inputs. Vanilla CNN as a ResNet architecture (He et al., 2016) is often limited by its local receptive field and lack of attention-based mechanisms, making it less effective in capturing global spatial relationships across modalities. It consists the backbone architecture for all ResNet-based baselines in this paper.

**SSVTP (Kerr et al., 2023a)**   The Self-Supervised Visuotactile Pre-training (SSVTP) framework employs contrastive learning with the InfoNCE loss to align visual and tactile representations. Visual data is treated as the query and tactile data as the key, aiming to minimize the distance between embeddings of matching pairs while maximizing the distance to non-matching pairs. This approach creates a shared visuo-tactile latent space by leveraging the spatial alignment of the data. However, SSVTP focuses solely on optimizing the visual-to-tactile direction.

**TAG (Yang et al., 2022)**   Building on the SSVTP framework, Touch-and-Go (TAG) employs a symmetric contrastive loss that aligns embeddings bidirectionally. In addition to optimizing visual-to-tactile alignment, TAG also trains the model to match tactile queries to visual keys. This dual-directional alignment combines both losses, ensuring the latent space captures strong bidirectional associations between the modalities. This enhanced alignment improves generalization across diverse material properties and multimodal applications, making TAG well-suited for tasks like tactile-driven image stylization and material recognition.

**MViTac (Dave et al., 2024)**   MViTac enhances TAG by integrating visual and tactile inputs through parallel ResNet-18-based encoders, using contrastive learning to align their representations in a shared latent space. Its dual strategy includes intra-modal contrastive learning for modality-specific consistency and

inter-modal contrastive learning for cross-modal alignment, both optimized with InfoNCE loss. A combined loss function balances these objectives, enabling robust multimodal representation and integration.

**UniTouch (Yang et al., 2024)**   UniTouch aligns tactile representations with pre-trained frozen vision-language models by leveraging a contrastive learning framework. It aligns tactile embeddings with pre-trained visual embeddings, which are already associated with other modalities like language and audio. This alignment is achieved using bidirectional contrastive objectives: tactile-to-vision and vision-to-tactile losses, combined into a unified loss function. By maximizing cosine similarity for paired visuo-tactile embeddings and minimizing it for unpaired ones, UniTouch creates a shared multimodal space for tactile, visual, and other modalities.

**SigLIP2 (Tschannen et al., 2025)**   SigLIP2 is a family of vision–language encoders trained on large-scale image–text pairs using a *sigmoid* contrastive objective. It extends this recipe with additional training components and data curation designed to improve transfer and robustness, yielding more capable image encoders than earlier SigLIP-style models. We use the released SigLIP2 base image encoder ($\sim$380 parameters) as a VLM-initialized baseline that has no tactile-aligned pretraining. To adapt SigLIP2 to our visuotactile setting, we treat tactile frames as standard RGB images and match the same input resolution and preprocessing used by our ViT-based baselines. We then fine-tune the resulting visuotactile model on the *source* dataset (TAG Category and, where applicable, OF-Real) using the same SSL optimization protocol as in Appendix B, and evaluate transfer with our standard *linear probing* and *zero-shot* protocols.

**VTT (Chen et al., 2022)**   The Visuotactile Transformer (VTT) introduces a transformer-based framework for visuotactile integration. By processing visual and tactile inputs through a shared encoder with cross-modal attention, VTT captures interdependencies between modalities. This approach enables the model to focus on critical task features by generating latent heatmap representations.

**RoPE (Heo et al., 2025)**   Relative Positional Encoding (RoPE) augments the transformer architecture with rotary positional encodings to capture relative spatial relationships within each modality. This method offers flexibility in sequence length and allows the model to capture decaying inter-token dependencies as their relative distances increase.

