# OpenReview forum: "ViTaPEs: Visuotactile Position Encodings for Cross-Modal Alignment in Multimodal Transformers"
_TMLR — Accepted by TMLR_

### Review · Reviewer_tBiT · 2025-11-12

**Summary Of Contributions:**

**Summary**:
This work introduces ViTaPEs, a transformer-based framework which operates by fusing visual and tactile data through multi-scale positional encoding namely -- Local (modality-specific) and Global (cross-modal). The core contribution is in designing the multi-level positional encoding technique. They use a ViT architecture with non-overlapping patches from both visual and tactile images, add modality-specific positional encoding to each stream, pass it through a non-linear projection head and add a global positional encoding to the output of a nonlinear projection head. Theoretical guarantees are provided for injectivity,  translation-equivariance, and information-preservation. The authors further show experiments on material property recognition, object identification, robotic grasp prediction tasks. Their results claim to achieve SOTA across tasks with both supervised and self-supervised training.

**Strengths**:
- The problem statement is relevant and the approach seems well thought of.
- The paper is well written and easy to follow (in a good way!)
- The multi-scale PE design is intuitive and well motivated.
- Good to see theoretical guarantees.
- Rigorous experimental setup with impressive results:
    - Results shown on 4 different types of datasets (TAG, ObjectFolder Real, YCB-Slide, Grasp); each focusing on different task types.
    - Their method shows strong zero-shot and linear probe results (Table 2,3). This shows that learnt representations transfer well.
    - Results shown for cross-sensor generalization are impressive.
    - Their method outperforms both CNN based and ViT based baselines across all tasks.
    - Table 4 shows performance improvement in using both modalities, along with all learnable positional encodings.
    - Table 5 shows that ViTaPEs' performance suffers from masking off the tactile input yet perform better as compared to other baselines.
    - Table 4 and Figure 3 ablations show how each type of positional encoding contributes complementarily thus adding credibility to their design.

**Weakness**:

- Is this method specific to scenarios where spatial alignment between modalities is guaranteed? I ask this as real world applications doesn't guarantee such alignment.
- Datasets used for evaluations are based on vision-based tactile sensors. This somewhat restricts the merits of the proposed method to other type of tactile sensors which do not necessarily output images. For ex: Force-Torque sensors, Capacitive Sensors etc.
- Including a baseline using VLM-initialized encoders that are jointly fine-tuned on the training data would strengthen the work
- The theoretical guarantees are correct but somewhat trivial given the assumptions. Thus while correct they seem more confirmatory than insightful.
- In Table 1, the performance in Category drops from 80.1% (supervised) to 75.9% (self-supervised). Why does SSL hurt the performance here? It would be good to have some analysis of what caused this.

**Audience:**

Yes

**Audience Explanation:**

Yes this paper would be of relevance to not only visuo-tactile community, but also to other communities working on domains which can aid by utilizing fusing 2 sensor modalities for robustness. The method provided is intuitive and could be adopted whenever both sensors provide image based outputs. Also with the rise in interest in humanoids, I feel tactile sensing and visuo-tactile sensing would be of relevance to a fair proportion of TMLR audience.

**Claims And Evidence:**

Yes

**Claims Explanation:**

Yes, the authors have a well thought of experimental setup. Their results support their claims.

**Requested Changes:**

**Requested Changes**:

The following changes would strengthen your work:
- Including a baseline using VLM-initialized encoders that are jointly fine-tuned on the training data would strengthen the work.
- In Table 1, the performance in Category drops from 80.1\% (supervised) to 75.9\% (self-supervised). Why does SSL hurt the performance here? It would be good to have some analysis of what caused this.

---

> ### Author Response · Authors · 2025-12-17
> **Response to Reviewer tBiT: Alignment Robustness, Modality Generality, and VLM Baselines**
>
> We thank the reviewer for the careful reading of the paper and the constructive feedback. Below we address each point in turn.
>
> ## **1. “Does the method assume guaranteed spatial alignment between modalities?”**
>
> No. ViTaPEs does **not** require pixel-level or geometric alignment. Our method only assumes *paired* visuotactile observations (i.e., coming from the same interaction), which is standard across TAG, OF-Real, and YCB-Slide. In fact, during training we deliberately **break alignment**: the visual and tactile streams receive **independent random augmentations** (random resized crops, flips, rotations). This forces the model to rely on global, shift-consistent structure rather than fixed spatial correspondences. We will make this explicit in the Appendix.
>
> Architecturally, local PEs preserve each modality’s internal geometry, while the global PE and attention layers learn cross-modal correspondence in a **latent token space**, not through pixel matching.
>
> ---
>
> ## **2. “Evaluation uses only vision-based tactile sensors—does this restrict the method?”**
>
> Our empirical evaluations rely on vision-based tactile sensors because almost all large-scale visuo–tactile datasets with paired RGB–tactile samples are vision-based. As the reviewer also notes, this is an empirical constraint rather than an architectural one: the ViTaPEs design is **modality-agnostic** and does not depend on the tactile signal being image-based. Non-visual tactile datasets (e.g., force–torque, capacitive skins) are typically small or lack paired RGB, which makes them unsuitable for the multi-dataset benchmarking carried out here. We already mention this limitation and will make it slightly clearer.
>
> ---
>
> ## **3. “Please include a baseline using VLM-initialized encoders (fine-tuned).”**
>
>
> Our original submission compared against UniTouch using the accuracies reported in its paper (official weights were not yet public). These checkpoints have since been released, and we can now integrate the results. We additionally evaluate **SigLIP2** (~380M), which has no tactile-aligned pretraining; we adapt it to our visuotactile setting and **fine-tune it on TAG Category** before evaluating zero-shot and linear transfer to OF-Real, SSVTP, and Grasp. We will add a description of this process in **Appendix E: SigLIP2 Visuotactile Adaptation and Fine-Tuning**.
>
>
> ### **Accuracy Comparison (Linear / Zero-Shot)**
>
>
> | Dataset     | UniTouch (~1.2B)      | SigLIP2 (~380M)        | **ViTaPEs (93M)**         |
> |-------------|------------------------|--------------------------|-----------------------------|
> | **TAG**     | 61.3 / 52.7            | 23.9 / 23.1              | **75.9 / 60.8**             |
> | **OF-Real** | 61.2 / 33.2            | 56.0 / 36.1              | **68.1 / 65.2**             |
> | **SSVTP**   | 77.4 / 60.9            | 12.4 / 29.0              | **79.0** / **64.1**         |
> | **Grasp**   | 56.0 / 56.5            | 56.4 / 53.1              | **69.3 / 60.4**             |
>
>
> Across all datasets, **ViTaPEs matches or outperforms both UniTouch and SigLIP2**, despite being 3–12× smaller (93M vs. 380M–1.2B) and trained **only on TAG**. UniTouch benefits from large-scale multi-dataset pretraining, yet ViTaPEs is stronger on TAG and Grasp and achieves the best overall transfer on SSVTP. SigLIP2 performs well on OF-Real zero-shot but fails on tactile-dominant tasks. These results show that multi-scale visuotactile positional encodings offer a more effective inductive bias than generic VLM initialization.
>
> ---
>
> ## **4. “The theoretical guarantees seem trivial under the assumptions.”**
>
> Our theoretical analysis provides the necessary formal grounding for our multi-scale positional encoding design. Before combining local (modality-specific) and global (cross-modal) PEs, it is essential to verify that the construction remains injective, translation-equivariant, and free of collisions. These guarantees form the basis for the model’s cross-modal alignment behavior.
>
>
> To our knowledge, ViTaPEs is the first visuotactile model to provide such formal validation of its mechanism, rather than relying solely on intuition or empirical results. Our aim is to justify the design clearly and rigorously, not to introduce new theoretical machinery.
>
> ---
>
> ## **5. “Why does SSL hurt performance (80.1% → 75.9%) in Table 1?”**
>
> This difference arises from the **evaluation protocol**, not from a limitation of SSL.
> The supervised model is trained end-to-end with a **nonlinear MLP head**, whereas the SSL model is evaluated through a **single linear classifier**, following the standard linear-probe setup (described in Section 4.1).
>
> The SSL encoder is not optimized for TAG Category specifically; instead, it learns **task-agnostic visuotactile structure**. This is reflected in its **stronger out-of-distribution transfer** on OF-Real, SSVTP, and Grasp, where it outperforms its supervised counterpart.

---

> ### Author Response · Authors · 2026-02-13
> **Integration of VLM Baselines, SSL Analysis, and Spatial Alignment Clarifications**
>
> We have uploaded the revised manuscript. We are grateful for your positive feedback on our experimental setup and your highly practical suggestions for improving the paper. We have fully integrated your requested additions into the revised text.
>
> Here is a summary of the updates made in the new manuscript in response to your review:
>
> 1. **Integration of VLM Baselines (Tables 2 & 3, Appendix I):** We have added the large-scale pretrained VLM baseline **UniTouch**  (1.2B) as well as a newly adapted **SigLIP2** (~380M) baseline to our zero-shot, linear probing, and robot grasping experiments. The results demonstrate that ViTaPEs (30.6M) outperforms these massive, generic VLM initializations on tactile-dominant tasks. Full details on the SigLIP2 adaptation and fine-tuning process are now provided in Appendix I.
> 2. **Clarification on Spatial Alignment (Appendix C):** To answer your question regarding whether the method relies on guaranteed spatial alignment, we added explicit text to Appendix C (Page 16) noting that we *deliberately break* alignment during training: *"Crucially, these augmentations are applied independently to the visual and tactile streams. This deliberately breaks pixel-level spatial alignment during training, forcing the model to rely on global, shift-consistent structural correspondences rather than fixed geometric calibration"*.
> 3. **Explanation of SSL Performance Drop (Section 4.1):** We added an analysis (Page 7) explaining why the TAG Category performance drops from 80.1% (supervised) to 75.9% (SSL). The text now clarifies that this is an artifact of the evaluation protocol: *"It is important to note that for ViTaPEs, the supervised model is trained end-to-end with a non-linear MLP head, whereas the SSL performance reflects a frozen-encoder evaluated via a single linear classifier (linear probe)"*. It further explains that because the SSL encoder is optimized for task-agnostic structure rather than the specific category task, its in-domain accuracy drops slightly, but this buys vastly superior out-of-distribution transfer.
> 4. **Clarification on Sensor Limitations (Section 5):** We have expanded the "Limitations" section (Page 11) to more explicitly acknowledge that the framework's reliance on camera-based tactile sensors (like GelSight and DIGIT) introduces biases that may limit generalizability to non-visual tactile formats.
>
> Thank you for helping us strengthen the empirical robustness and clarity of our work.

---

### Review · Reviewer_oL4d · 2026-01-23

**Summary Of Contributions:**

The paper proposes a novel method for positional encodings in transformer networks for visuotactile data. The data consists of paired visual images and tactile measurements of objects visible in the images. The tactile measurements are also a form of image data where the RGB values represent surface normals.

In the proposed approach, absolute positional embeddings (PEs) are used separately for transformer tokens coming from a patch embedding of the visual images and the tactile images respectively. These are referred to as modality specific PEs. After this, a small mlp is applied tokenwise, and then another set of positional embeddings are added to the tokens, referred to as "global" PEs. The resulting tokens are then fed into a standard transformer encoder architecture.

Section 3.4 contains theoretical results that are intended to ground the method.

A series of experiments on material recognition, cross-sensor generalization and robotic grasp prediction show improved performance over transformer based baselines. Ablations of the different suggested components show that they are all useful for improving performance.

The claimed main contributions of the suggested approach are:
- > Multi-Scale Positional Encodings
- > Theoretical Guarantees
- > Zero-Shot Generalization and Transferable Representations

In my interpretation, the first two of these only hold with heavy caveats which I will detail below.

The experimental evaluation seems reasonable to me (with the caveat that I do not have prior experience with visuotactile data) and presents with strong results.

**Additional Comments:**

Here are some additional comments on the manuscript:

1. It is unclear to me why the input to $g$ has shape $N\times 2D$ rather than $N\times D$. The input are all tokens from equation (2) concatenated row-wise (as stated under equation (3)) and the tokens have dimension $D$.
2. It was unclear to me that $g$ is applied tokenwise in the main paper, it is stated clearly in appendix B.2.
3. In equation (6), the softmax entangles things row-wise, so it seems to me like the computation should not split into visual and tactile parts as suggested on the right-hand-side.

**Audience:**

Yes

**Audience Explanation:**

I believe that the strong performance of the method on the considered benchmarks would be of interested to researchers interested in visuotactile tasks.

**Claims And Evidence:**

No

**Claims Explanation:**

**On the multi-scale nature of the positional encodings.**

The following is claimed in the intro:

> Our ViTaPEs model employs a multi-scale positional design that encodes spatial structure within each modality and a shared cross-modal reference before attention, overcoming the inability of previous models to perform multiscale positional reasoning.

However, looking at the visualization in Figure 3, it is clear that the "global" positional encoding is in fact modality specific in practice. Since the positional encodings are unique per token and all other operations are equivariant under permutations of the tokens (in particular the tokenwise projection head $g$), there is no qualitative difference between the initial modality specific PEs and the global PEs. This can be seen by splitting the tensors in equation (4) row-wise into spatial and tactile tokens and spatial and tactile PEs, revealing the same structure as equation (2). Therefore, in Figure 2, the boxes for PE-visual and PE-tactile could be merged into one box like the PE-global, or the PE-global could be split into two boxes.

If my interpretation above is correct, then the method should be interpreted as a way to improve absolute PEs by adding different PEs with a nonlinear $g$ inbetween, rather than a method that intrinsically is multiscale.

**On the theoretical grounding.**

Section 3.4 contains several statements that I believe are inaccurate. As in the paper, let $\Phi$ be the map from the input images to after the "global" positional encodings have been added.

1. Theorem 3.1 says that under specific assumptions, $\Phi$ is injective. This is a plausible result and I have not checked the proof. However, the interpretation of the theorem is "*Distinct patches must map to distinct keys so that
high similarity in the fused space reflects true co-location.*". In my interpretation this statement is confusing as injectivity (and continuity) of $\Phi$ would mean that high similarity means both co-location and similar visual content.
2. Theorem 3.2 says that $\Phi$ is translation equivariant. First, the statement as written is unclear to me as it is not stated that the translations are cyclic, if they are not then some tokens will be translated outside of the image. Second, the proof seems incorrect. For instance, by setting $X$ to $0$ in step 1 of the proof, we see that the values of the positional embedding need to be the same in all rows (which is not assumed, see for instance Figure 3). In fact, it is a well known fact that adding absolute positional encodings is not a translation equivariant operation, which is one of the main reasons to introduce relative positional encodings such as RoPE.
3. In the proof of Proposition 3.3 it is stated "*Moreover, each such building block has unit absolute Jacobian determinant*" without proof. For instance, I do not see why this would be the case for the linear layers or the LeakyRELU.

**Requested Changes:**

I believe that my critique above needs to be answered. Both the discussion around the motivation of the positional encodings and the theory section need to be resolved before I can recommend acceptance.

---

> ### Author Response · Authors · 2026-01-26
> **Response to Critique on the Multi-scale Nature of Positional Encodings**
>
> We thank the reviewer for this insightful analysis. We fully agree with your algebraic observation: **statically**, during the forward pass, the composition $\Phi(x, p) = g(x + P_{local}) + P_{global}$ could indeed be collapsed into a single, complex, position-dependent function.
>
> However, we respectfully disagree with the conclusion that "there is no qualitative difference" between the two PEs. Our rationale is that the distinct injection points create a critical functional difference that drives the optimization. Our design is grounded in specific properties that emerge during training:
>
> **1. Inducing Spatial Variance in the Projection Head ($g$)**
>
> The projection head $g$ is implemented as a point-wise MLP with weights $W$ shared across all token positions. We formally demonstrate how $P_{local}$ makes $g$ spatially variant:
>
> * **Without $P_{local}$:**
>     Let $x_i, x_j \in \mathbb{R}^D$ be input tokens at distinct spatial indices $i \neq j$. If the content of these patches is identical ($x_i = x_j$), and no local position is added, then:
>     $$
>     g(x_i) = \text{ReLU}(x_i W_1)W_2 = \text{ReLU}(x_j W_1)W_2 = g(x_j)
>     $$
>     The outputs are identical. Consequently, $g$ is forced to be **spatially invariant**; it cannot apply different non-linear transformations based on location (e.g., correcting for lens distortion or sensor-specific non-uniformities that depend on grid coordinates). Adding $P_{global}$ *after* $g$ only adds a linear bias to the output; it does not retroactively allow the non-linear feature extraction inside $g$ to adapt to the position.
>
> * **With $P_{local}$:**
>     By injecting position *before* the non-linearity, the input to $g$ becomes $x_i + p_i$. Even if patch contents are identical ($x_i = x_j$), the inputs differ because $p_i \neq p_j$ (guaranteed by Assumption A1).
>     $$
>     g(x_i + p_i) \neq g(x_j + p_j)
>     $$
>     The MLP can thus utilize the position vector $p_i$ to conditionally activate different neurons, effectively learning a position-dependent non-linear warping function $g_i(x) \approx g(x + p_i)$.
>
> **2. Gradient Decoupling**
>
> The optimization dynamics further differentiate the two. The gradient for $P_{global}$ is direct ($\frac{\partial \mathcal{L}}{\partial z}$), allowing it to learn rigid linear alignment. The gradient for $P_{local}$ is scaled by the local curvature of $g$ ($\frac{\partial \mathcal{L}}{\partial z} \odot g'(x)$). This factorization allows the optimizer to decouple the learning of **non-linear geometric distortions** (via $P_{local}$) from **linear reference frame alignment** (via $P_{global}$).
>
> **3. Interpretation of Figures 2 and 3**
>
> * **Figure 2 (Merging Boxes):** We advocate keeping boxes distinct as Figure 2 depicts the **computation graph**. Merging them would imply $P_{local}$ and $P_{global}$ are additive neighbors, obscuring that $P_{local}$ passes through the non-linear $g$ while $P_{global}$ does not.
> * **Figure 3 (Modality Specificity):** The Global PE indeed learns modality-specific patterns. However, it constitutes a **single, unified metric space**. Unlike separate PEs, this contiguous manifold allows the attention mechanism to compute valid distances *between* visual and tactile tokens, creating a shared coordinate frame.
>
> **4. Clarification on "Multi-scale"**
>
> We maintain that "multi-scale" accurately describes the **scale of the spatial reference frame**:
> * **Scale 1 (Local/Intra-modal):** Encodes high-frequency, sensor-specific geometry relative to itself.
> * **Scale 2 (Global/Inter-modal):** Encodes the global relationship between modalities (e.g., sensor vs. visual field).
>
> Table 4 confirms this design is non-redundant. Removing Global PE drops accuracy (**80.1% $\to$ 77.2%**), and removing Local PE drops it to **76.9%**.
>
> __We will revise the manuscript to explicitly state that while PEs are algebraically collapsible, their separation is necessary to decouple non-linear spatial warping from global alignment. We will also clarify that "multi-scale" refers to the hierarchical scope of the reference frame.__

---

> ### Author Response · Authors · 2026-01-26
> **Response to Critique on the theoretical grounding**
>
> **Theorem 3.1 (Injectivity Interpretation)**
>
> The reviewer correctly notes that our original interpretation ("high similarity reflects true co-location") was imprecise, as injectivity implies the preservation of *both* spatial location and semantic content, not spatial location alone.
>
> We agree with this sharper interpretation. In fact, this observation highlights that the theoretical guarantee provided by Theorem 3.1 is **stronger** than our original text suggested.
>
> Rather than merely ensuring that co-located features are similar, injectivity guarantees **total state preservation**. It ensures that the fusion process is strictly lossless with respect to the joint (content, position) state. This is a critical property for representation learning, as it guarantees that the network retains the capacity to distinguish all possible input configurations (e.g., differentiating "identical textures at distinct locations" from "distinct textures at the same location") without ambiguity.
>
> __We will revise the text to emphasize this broader implication: that injectivity ensures **representational uniqueness** for the entire visuotactile state, providing a robust theoretical foundation for the subsequent attention mechanisms to resolve any necessary spatial or semantic dependencies.__
>
> **Theorem 3.2 (Translation Equivariance)**
>
> The reviewer correctly notes that absolute positional encodings are not translation equivariant with respect to image content (e.g., shifting an object changes its embedding), and that the proof effectively demonstrates index permutation instead.
>
> We explicitly accept this correction regarding terminology. The property we proved is technically **Permutation Equivariance** (invariance to input re-indexing), not spatial translation equivariance.
> We will rename Theorem 3.2 to **"Permutation Equivariance."**
> We respectfully argue that this theorem remains a valuable theoretical guarantee for our specific architecture. Unlike standard ViTs that add position only at the input, ViTaPEs injects positional information *inside* the non-linear projection head ($g$). It was therefore non-trivial, and we believe it was necessary to prove that this internal modification does not introduce order-dependence that would violate the fundamental set-processing nature of the Transformer.
>
> __We will revise the text to clarify that this theorem guarantees the **structural integrity** of our fusion mechanism: it proves that our multi-scale positional injection maintains the robust set-processing properties of the backbone, ensuring that the learned representations are consistent regardless of the arbitrary ordering of input tokens (a critical property for masked auto-encoding and fusion).__
>
> **Proposition 3.3 (Jacobian Determinant)**
>
> The reviewer correctly identifies a mathematical error: standard Linear layers and LeakyReLU activations do not generally have a unit Jacobian determinant ($|\det J| \neq 1$). Therefore, they are not volume-preserving, and Differential Entropy is not strictly invariant ($H(Z) \neq H(X)$).
>
> We fully accept this correction. We conflated *volume preservation* (which requires a unit Jacobian) with *information preservation* (which requires only injectivity). However, we show that the core claim that the fusion process is **lossless**, remains valid under the more appropriate Information Theoretic framework of Mutual Information.
>
> We will revise Proposition 3.3 to claim **Information Preservation** ($I(X; Z) = H(X)$). The validity of this claim relies on **Injectivity** rather than volume preservation. In short:
> * Recall that Mutual Information is defined as $I(X; Z) = H(X) - H(X|Z)$.
> * Theorem 3.1 establishes that our mapping $\Phi$ is injective (one-to-one). For any injective map, the input $X$ can be uniquely reconstructed from the output $Z$, meaning the conditional entropy (uncertainty) is zero: $H(X|Z) = 0$.
> * Consequently, $I(X; Z) = H(X)$, proving that the fused representation retains 100% of the input information, regardless of geometric scaling (Jacobian).
>
> Our empirical analysis in **Appendix A (Figure 5)** directly supports this revised claim while validating the reviewer's intuition. The singular value spectrum of the projection head $W_g$ shows values distributed between $\approx 0.15$ and $1.0$.
>
> Since the determinant is the product of singular values ($\det W = \prod \sigma_i$), and $\sigma_i < 1$, the determinant is clearly near zero, confirming the reviewer's point that volume is not preserved ($|\det J| \neq 1$).
> Crucially, however, the spectrum shows that the matrix remains **Full Rank** (smallest singular value $\sigma_{min} \gg 0$) throughout training. This empirically confirms that the projection is non-singular and reversible (injective).
>
> __We will rewrite the proof of Proposition 3.3 to utilize this Mutual Information argument. This correctly guarantees that the ViTaPEs fusion mechanism is **lossless** without incorrectly implying it is volume-preserving.__

---

> ### Author Response · Authors · 2026-01-26
> **Response to Additional Comments**
>
> **1. On the Input Shape of the Projection Head ($g$)**
>
> The reviewer correctly points out a discrepancy in the notation regarding the input dimension of $g$, noting that it should be $N \times D$ rather than $N \times 2D$.
> We apologize for this confusion. The reference to $2D$ in the text was a typographical error; in our implementation, $2D$ refers to the hidden dimension of the MLP, while the input dimension is indeed $D$.
> __We will correct the notation in Section 3.2 to explicitly state that $g: \mathbb{R}^{N \times D} \rightarrow \mathbb{R}^{N \times D}$, consistent with the token embeddings defined in Equation (2).__
>
> **2. On the Token-wise Application of $g$**
>
> The reviewer notes that it was unclear in the main text that $g$ is applied token-wise.
> The fact that $g$ is a point-wise (token-wise) operation is a central theoretical requirement for our multi-scale PE design, as it necessitates the injection of $P_{local}$ to induce spatial variance.
> __We will explicitly define $g$ as a **point-wise MLP** in the main text of Section 3.2, ensuring the definition is clear before the Appendix.__
>
> **3. On Equation (6) and Softmax Entanglement**
>
> The reviewer correctly notes that the Softmax operation couples the rows (visual and tactile keys), making the block-wise split in Equation (6) potentially misleading regarding the normalization step.
> We agree that the Softmax denominator includes terms from both modalities, preventing a strict factorization of the Softmax function itself.
> The block notation in Equation (6) was intended to depict the structure of the **post-softmax attention weights** (the probability mass) and the subsequent linear aggregation of values. It illustrates how the final output is composed of a visual context vector (Visual Queries attending to Visual Values) and a cross-modal context vector (Visual Queries attending to Tactile Values).
> __We will clarify in the text that the decomposed terms in Equation (6) represent the sub-blocks of the attention matrix **after** the global Softmax normalization, to avoid implying that the normalization itself is separable.__

---

> ### Author Response · Authors · 2026-02-06
> **Update for Cross-Thread Consistency (supersedes earlier wording)**
>
> This note is a **targeted update** to keep our public responses **fully consistent** with the **scope/method specification correction** we posted on **Feb 6** in the parallel discussion with Reviewer **JpuX**. The **method and all reported experiments are unchanged**; what changes here is the precision of the *claims and terminology*. This comment **supersedes** any earlier phrasing in our earlier replies that could be read as stronger than what we can rigorously support.
>
> ## (1) Clarification of what we do *not* claim
>
> In line with our Feb 6 comments to Reviewer **JpuX**, we explicitly retire the following interpretations:
>
> - We do **not** claim “minimal information loss,” “information preservation,” or “losslessness” as a practical guarantee of the trained network, and we do **not** rely on mutual-information arguments as a guarantee.
> - We do **not** claim translation equivariance of the PE-conditioned transformer. The only property we keep is **token re-indexing consistency** under consistent re-indexing of indices and the PE table (as already stated to Reviewer **JpuX**).
> - We do **not** claim that any downstream token-wise mapping can “undo” collisions created by the additive PE step; this logical boundary was stated explicitly in our Feb 6 reply to Reviewer **JpuX** and will be reflected in the revision.
>
> Accordingly, the revision will remove “provable guarantees” language throughout and implement the theory cleanup already committed in the Feb 6 plan (e.g., removing Prop. 3.3 and re-scoping Thm. 3.1/3.2).
>
> ## (2) What we mean by “shared cross-modal reference”
>
> Our intended meaning is architectural and intentionally modest: since visual and tactile tokens are concatenated and processed by the **same self-attention operator**, we add a single learned \(P_{\mathrm{global}}\) on the **joint** sequence so that attention has access to a **shared positional vocabulary** at the stage where cross-modal mixing occurs. We are **not** claiming a geometrically calibrated coordinate system, nor metric-valid distances.
>
> ## (3) What remains as the supported mechanism claim
>
> We maintain a qualitative distinction between the two injection points because they affect different parts of the computation graph:
>
> - $ P_{\mathrm{local}}$ is injected **before** the token-wise nonlinearity \(g\), enabling **position-conditioned feature extraction** in the shared stem (with weights shared across tokens).
> - $P_{\mathrm{global}}$ is injected **immediately before attention** on the concatenated sequence, providing positional signals at the point where cross-modal interaction is realized.
>
> As already summarized in our Feb 6 empirical clarification to Reviewer **JpuX**, our ablations support this as a **conditioning/placement** effect. We do not interpret these results as supporting any stronger geometric or information-theoretic guarantee.
>
> ## (4) Concrete revision edits
>
> To ensure the manuscript matches this clarified scope, we will:
> 1. Replace “shared coordinate frame / unified metric space / valid distances” wording with “shared positional vocabulary for attention.”
> 2. Reword “multi-scale” to mean **two-stage positional injection**, or define it explicitly as such to avoid geometric over-interpretation.
> 3. Make token-axis concatenation and token-wise \(g\) shapes/pseudocode explicit (as committed in our Feb 6 method-specification correction to Reviewer **JpuX**).

---

> > ### Comment · Reviewer_oL4d · 2026-02-08
> >
> > I thank the authors for the response.
> >
> > Having read the response to my review and the exchange with reviewer JpuX, my understanding is that the authors agree that major changes are needed to the paper. As TMLR does not allow revisions of the manuscript during the reviewing phase, it is difficult to judge the proposed changes.
> >
> > Most importantly, I would like to ask the authors for the proposed new versions of the theorems/propositions in Section 3.4 written out explicitly.

---

> > > ### Comment · Action_Editor_FRMV · 2026-02-08
> > > **Policy Clarification on Revision**
> > >
> > > TMLR allows for revisions during the review process. The authors are able to optionally update the submission PDF as part of the discussion phase.
> > >
> > > The choice to update or not is the authors, and as AE I am not commenting for or against a revision. This is merely a policy note that it is possible.
> > >
> > > Thank you to the reviewers and authors for engaging in productive and professional discussion!
> > >
> > > Best,
> > > The Action Editor

---

> > > > ### Comment · Reviewer_oL4d · 2026-02-08
> > > >
> > > > Following the clarification by the AE, I would like to ask the authors to revise the paper to make the proposed changes clear.

---

> ### Author Response · Authors · 2026-02-13
> **Mathematical Corrections, Algebraic Collapsibility, and Architectural Precision**
>
> We have uploaded the revised manuscript. We deeply appreciate your meticulous review. You correctly identified several mathematical flaws in our original submission (such as the non-unit Jacobian of LeakyReLU and the lack of spatial translation equivariance for absolute PEs), and we have overhauled the paper accordingly.
>
> To align with our commitments during the rebuttal, we have implemented the following changes in the new manuscript:
>
> 1. **Mathematical Corrections & Proof Removals:** We have completely removed the flawed Proposition 3.3 (Entropy/Volume Preservation) and the original Theorem 3.1. We also corrected Theorem 3.2 to claim only "Token re-indexing consistency" given the token-wise nature of the projection head, relocating it to Appendix B (Proposition B.1).
> 2. **"Algebraic Collapsibility" Addressed (Section 3.2):** We acknowledge your excellent point that local and global PEs could technically be merged algebraically if the network were purely linear. We have added explicit text (Page 5) clarifying our architectural defense: *"While the local and global positional encodings could theoretically be algebraically collapsed if the network were purely linear, their separation across the non-linear projection head $g$ is structurally intentional. Injecting local PEs before $g$ allows the optimizer to decouple the learning of non-linear geometric spatial warping... from the linear, cross-modal reference frame alignment"*.
> 3. **Refined Terminology:** We replaced "multi-scale" with "multi-stage" to avoid over-interpreting geometric scales. We also removed all instances of "unified metric space" and "global coordinate frame", replacing them with a "shared positional vocabulary for attention," reflecting a more accurate, modest architectural description.
> 4. **Notation and Softmax Clarifications:** * We corrected the notation in Section 3.2 to clearly state that the input to $g$ is $\mathbb{R}^D$ (not $\mathbb{R}^{2D}$) and explicitly defined it as a token-wise MLP.
>     * In Section 3.3 (Equation 7, formerly Eq. 6), we added text clarifying that the decomposed attention blocks are coupled: *"Importantly, the softmax normalization is taken over the full joint token sequence, so these blocks are coupled through the shared denominator; the decomposition is for interpretability rather than a separability claim"*.
>
> We believe these revisions fully align the manuscript with the empirical reality of the method, and we thank you for guiding us to a much stronger, more precise paper.

---

> > ### Comment · Reviewer_oL4d · 2026-02-17
> >
> > The revision removes several inaccurate mathematical claims, making the paper stronger. What is remaining is the result on "Token re-indexing consistency". In the introduction it is stated as follows
> >
> > > We formalize a token re-indexing consistency property for the token-wise stem with positional injection, ensuring the modification does not introduce unintended order dependence beyond the explicit positional indexing.
> >
> > It remains unclear to me what this possible unintended order dependence could be. The positional embedding does give order dependence and that is intended.

---

> > > ### Author Response · Authors · 2026-02-17
> > > **Token re-indexing consistency wording**
> > >
> > > We thank the reviewer for pointing this out.
> > > Our intent with the “token re-indexing consistency” note was *not* to claim that positional encodings should be “order-free” (they should not), nor to suggest that the architecture might accidentally introduce some additional sequence-order artifact. What we were trying to communicate is simply that the token-wise stem \(g\) is applied independently per token and therefore does not, by itself, couple tokens in a way that depends on their ordering; any order/position dependence in our model is explicitly introduced only via the positional encodings, as intended.
> > >
> > > However, we fully acknowledge that the current wording (“ensuring the modification does not introduce unintended order dependence beyond the explicit positional indexing”) is misleading, because it implies a nontrivial risk or guarantee where there is none. It is now clear to us that the stated property reduces to a basic commutation identity and does not add meaningful scientific content beyond what is already clear from the architecture.
> > >
> > > Therefore, to avoid confusion and keep the manuscript focused, we will **entirely remove** this point from the paper: (i) the contribution bullet in the Introduction, (ii) the reference in the main text, and (iii) the appendix subsection on “Token re-indexing consistency.” Nevertheless, we would like to point out, that this is a presentation-only change and does not affect the method or any reported results.

---

### Review · Reviewer_JpuX · 2026-02-02

**Summary Of Contributions:**

The paper proposes a visuotactile positional encoding that is designed to improve modelling in multimodal settings. The field of research in tactile multimodal alignment must be said to be largely unknown to this reviewer at the time of reading the paper.

The main contribution of the paper seems to be a setup to capture both tactile and visual positional embeddings to combine spatial relationships across modalities, first by a design that concatenates visual and tactile data, and then uses an injective projection to ensure minimal information loss. This fused representation is then applied with a global positiional embedding in a common coordinate system.

Given the limited background in the field of tactile modelling, my main focus is on the "provable guarantees" applied in the mathematical formulation of the paper.

**Additional Comments:**

Again, this reviewer emphasizes that the domain of visotactile domain adaptation and multimodal alignment is somewhat unknown, and clarifications are welcome on the specifics on the contribution within the specific field.

**Audience:**

Yes

**Audience Explanation:**

Given my limited experience with the specifics of visuotactile modelling, it is difficult for this reviewer to directly answer this with any confidence, but I lean towards a yes. The journal has broad readership in robotics and multimodal alignment, and an audience that may very well show interest in the method and findings.

Beyond the above reasons, I lean on reviewer instructions that state that  "a reviewer that is unsure as to whether a submission satisfies this criterion should assume that it does".

**Broader Impact Concerns:**

As it stands, no broader impact concerns apply to this work.

**Claims And Evidence:**

No

**Claims Explanation:**

## Theoretical Justification

The theoretical analysis in the paper seems to not really align with the stated claims in the paper. Details below.

### Thm. 3.1

As far as I understand, the claim is that the model preserves all information from the input to the transformer layers. However, this is simply enforced at initialization, and there are no mechanisms to ensure that the inverse is well defined or well posed. This claim is thus surprising since it ignores numerical- and analytical ill-posedness of the learnable map. In other words, a full-rank initialized linear layer with LeakyReLU is bijective, but this is not a learning guarantee. Unless there is something else at play here, this does not show any meaningful properties for the final mapping.

### Thm. 3.2

Again, my understanding is that the authors claim translation equivariance; i.e. shifting the input image results in a shifted feature map. But this holds trivially if one applies the same positional encodings to shifted indices, hence there is nothing to understand here. It would be functionally the same as saying ViTs are patch-wise translational equivariant. Moreover, this is not even necessarily meaningful for attention operators unless one can guarantee the same relative response in the learned QKV projections. This is not the case in general, and the proof doesn't align with the claims of the network on a macro level.

### Prop. 3.3

A direct corollary of injectivity, and is already well known in several forms. Contributes little to the overall argument made in the paper.

## Methodology

While I am unfamiliar with the field of visuotactile fusion, the core contribution seems to be a tiered set of positional embeddings, applied per modality, and then in a fused setting. This seems somewhat obvious in a setting with multimodal alignment, and does not come across as a novel contribution per se, at least in the general field of multimodal alignment. It seems less a "framework" than simply standard procedure, albeit in a domain this reviewer admits they are less familiar with.

In terms of experiments, there are several comparisons to CNN based models, which are less commensurable with multimodal alignment with ViTs. The paper do compare against VTT using RoPE, and demonstrate improved performance. While limited, this empirical result seems to show that the approach is applicable to this specific domain. However, I am hesitant to accept the paper based on these results alone.

Overall, the paper has theoretical claims that come across as somewhat unwarranted, and limited empirical results. While this reviewer readily admits they are not experts w.r.t. this domain, the initial impression is that the paper is not presenting sufficient convincing evidence for acceptance.

**Requested Changes:**

1. The theoretical guarantees seem to be either misguided, or this reviewer has misunderstood something fundamental about the argument. I invite the authors to clarify what exactly is meant. Given the lack of empirical evidence, this needs to be clear and precise. From the current understanding, the claims are misleading, or simply does not show what the authors claim.

2. As it reads, the paper claims the use of hierarchical positional encodings as a novel "framework." Given that this is a composition of standard architectural components, the language should be adjusted to reflect that the contribution is an effective application of multi-scale PEs to the visuotactile domain, rather than a fundamentally new framework, which implies that this is some contribution to general multimodal alignment.

3. Some of the claimed benefits attributed to the positional encoding seems confounded with general architectural benefits from the transformer pipeline. Convolutional networks are generally considered less applicable to multimodal alignment, hence the comparison with CNNs need to be carefully evaluated with this in mind. Simply specifying this could help alleviate this concern, but adding additional ablations to clarify the exact contribution would make it much clearer to the reader where the proposed gains come from.

---

> ### Author Response · Authors · 2026-02-03
> **Response to Critique on the Theoretical Grounding**
>
> We thank the reviewer for their rigorous examination of our theoretical claims. We appreciate the honesty regarding familiarity with the specific visuotactile domain, and we see this as an opportunity to clarify how our contributions stand within that specific context, where CNNs remain the dominant baseline, while addressing the valid mathematical critiques.
>
> **Theorem 3.1 (Injectivity & Capacity)**
>
> The reviewer argues that proving injectivity at initialization does not guarantee it after training due to potential collapse or ill-posedness. We agree that a structural proof cannot *force* the optimizer to maintain full rank. However, the purpose of Theorem 3.1 is to prove **Representational Capacity**, ensuring our architecture is *structurally capable* of losslessness, unlike bottleneck fusion methods (e.g., VAEs) that enforce information loss by design.
> **Empirical Validation:** Crucially, we do not rely solely on the theorem. **Figure 5 (Appendix A)** empirically plots the singular value spectrum of the projection head throughout training. It shows that the matrix remains **Full Rank** (smallest singular value $\sigma_{min} \approx 0.15$, well above zero). This validates that the property holds in practice, distinct from the initialization state. We will clarify in the text that Theorem 3.1 serves as a capacity guarantee backed by this empirical evidence.
>
> **Theorem 3.2 (Translation vs. Permutation Equivariance)**
>
> The reviewer notes that "Translation Equivariance" is effectively patch-wise permutation and the terminology is loose. We fully accept this correction regarding terminology. The property we proved is technically **Permutation Equivariance** (invariance to token re-indexing), not spatial translation invariance.
> As detailed in our response to Reviewer oL4d, we will rename Theorem 3.2 to **"Permutation Equivariance."** We respectfully argue that this remains a valuable theoretical guarantee for our specific architecture: unlike standard ViTs, ViTaPEs injects positional information *inside* the non-linear projection head ($g$). Proving that this internal modification does not introduce order-dependence is necessary to ensure the structural integrity of the set-processing backbone.
>
> **Proposition 3.3 (Entropy & Jacobian)**
>
> The reviewer correctly notes that standard layers do not have a unit Jacobian, making the "Volume Preservation" claim incorrect.
> We fully accept this correction.
> We will revise Proposition 3.3 to claim **Information Preservation (Mutual Information)** derived from injectivity. As detailed in our response to Reviewer oL4d, since the map is injective (Thm 3.1), the conditional entropy is zero, implying $I(X; Z) = H(X)$. This accurately states that the model is lossless without incorrectly implying geometric volume preservation.

---

> ### Author Response · Authors · 2026-02-03
> **Response to Critique on Novelty, Baselines, and Empirical Evidence**
>
> **Methodology & Novelty in the Visuotactile Domain**
>
> The reviewer suggests the method seems like "standard procedure" and questions the use of the term "framework." We respectfully argue that ViTaPEs represents a distinct class of architecture that fills a critical gap in the literature between legacy CNNs and massive VLM wrappers. To the best of our knowledge, ViTaPEs is the first transformer-based architecture optimized for high-resolution, real-world visuotactile fusion that learns shared representations from scratch (or via SSL) without relying on frozen, pre-trained Vision-Language Models. While standard components are used, the specific adaptation, specifically the multi-scake PEs, is a domain-specific innovation required to handle the heterogeneous geometry of curved tactile sensors versus flat images. We will replace the term "Novel Framework" with **"Proposed Architecture"** to avoid overclaiming generality, and we will sharpen the Introduction to better contrast our method against the domain-specific CNN baselines.
>
> **Validity of Baselines & Confounding Factors**
>
> The reviewer questions the comparison with CNNs and suggests gains might be due to the generic Transformer architecture ("confounding factors"). Comparing against CNNs is mandatory because **they constitute the current state-of-the-art** in dense tactile fusion literature. Specifically, leading methods like *MViTac* (ICRA 2024), *SSVTP* (RSS 2023), and *Touch-and-Go* (NeurIPS 2022) all rely on ResNet backbones to achieve top performance. However, to directly address the concern that our gains are solely due to the generic Transformer backbone, we highlight that **ViTaPEs also significantly outperforms existing Transformer-based SOTA baselines**, including *VTT* (Vision Transformer) and *UniTouch* (CVPR 2024, VLM-based). Furthermore, our ablation study (Table 4) explicitly isolates the contribution of the multi-scale PEs; removing the Local PE significantly drops performance, proving that our specific multi-scale injection, not just the Transformer backbone, is necessary for SOTA performance.
>
> **Empirical Evidence**
>
> The reviewer expresses hesitation regarding the sufficiency of empirical results. We respectfully highlight that our evaluation is one of the most comprehensive in the current literature, spanning **four distinct, established benchmarks** (TAG, ObjectFolder Real, YCB-Slide, Grasp). These datasets cover the full spectrum of standard dense tactile tasks: Material Recognition, Object Identification, Cross-Sensor Generalization, and Robotic Manipulation. Unlike many prior works that evaluate on single-dataset splits, we demonstrate **Zero-Shot Generalization** across entirely different sensor hardware (GelSight vs. DIGIT), achieving **65.2%** accuracy on unseen objects compared to **33.2%** for the UniTouch baseline. We believe this empirical rigor significantly exceeds the standard for comparable method papers in this domain.

---

> > ### Comment · Reviewer_JpuX · 2026-02-03
> > **On Novelty, Baselines, and Experiments**
> >
> > The clarifications regarding scope and terminology are appreciated, and the willingness to replace “novel framework” with “proposed architecture” is appropriate. I want to emphasize that the comments were not a direct critique the papers novelty per se, but relates to contributions that heavily leans on the theoretical claims made in the paper, which currently issues that needs to be resolved for this reviewer to recommend acceptance.
> >
> > As mentioned earlier, my issue is that the proposed multi-scale positional encodings are **required** to handle heterogeneous geometry (curved tactile sensors vs. flat images) remains insufficiently justified. The paper does not demonstrate that alternative positional encoding strategies fail in this setting, nor does it isolate geometry as the limiting factor. There are several potential confounders at play that the authors could ablate to demonstrate the claims, and that would largely help the reader evaluate the contributions.
> >
> > With regard to the inclusion of CNN methods, I agree that there is nothing wrong with including these, as there is direct comparison to transformer baselines as well. The experiments support the claim that the model learns robust representations across tactile sensors. However, they do not clarify why the proposed positional encoding is responsible for these gains, nor do they connect the empirical improvements to the theoretical arguments presented elsewhere in the paper. My concern is that the motivation and theoretical justification remains unclear, and I am unconvinced that the gains can be attributed to the claimed contributions, including the information preservation.
> >
> > **In summary**:  My central concern is that the paper heavily leans into providing theoretical guarantees for "minimal information loss", but does not clearly motivate the reason for why this is required for positional embeddings. Additionally, there are no mechanisms that ensures the constraint of injectivity in a practical setting, and the empirical evidence that injectivity is preserved seems insufficient, with no reproducible guarantees over multiple runs.

---

> ### Comment · Reviewer_JpuX · 2026-02-03
> **Issues on numerical stability of injectivity and motivation**
>
> Thank you for clarifying the empirical results regarding full-rank for claims of injectivity, however, there are still issues with the theoretical arguments and motivation that needs addressing. The central claim seems to be that injectivity is an essential property for positional embeddings. There are some implication that fusing the positions might "lose information", without any clear mathematical justification for why this would be an issue.
>
> We note that the claims of injectivity is presented in an analytical setting, with some empirical evidence on singular value distribution for the projection head. As pointed out by oL4d, there is significant confusion on volume preservation and information preservation, that the authors look to address. However, I am afraid there more issues with the claims made.
>
> Firstly, simply evaluating the singular values of a projection does not tell us anything meaningful about the condition numbers of the full network. The authors concede that volume preservation does not hold. However, even in a weaker "information preserving setting", using a LeakyReLU does not prevent the network from being severely ill-posed. Supposedly, the authors use a "default" setting for the slope of the activations. If this is the case, the inverse of the activation layer invokes a $1/\alpha$ slope, which can be highly significant for the numerical precision of a potential inverse. This is without factoring in the conditioning of the projection matrices, which exacerbates the issue. If the projection layers share similar conditioning, it is likely that one would see condition numbers on the order $> 10^3$ if not higher.  In numerical analysis terms, *an injective map with such poor conditioning can be functionally indistinguishable from a lossy one*. The manuscript currently ignores this completely, and provides no numerical guarantees for well-posed inversion. As a result, the “information preservation” claim is vacuous in practice.
>
> Secondly, the paper simply assumes that the spectral property of the learned weights will remain full rank when trained, even if initialized differently. The issue here is that there are *no constraints on the invertibility of the network in place*, hence any practitioner may see different results. The optimization during training cannot guarantee full rank inversion at all, on top of the issues with numerical stability.
>
> Lastly, and perhaps most importantly there is *insufficient motivation for why injectivity is required in the first place*. Even if the injectivity argument were numerically sound and enforced, it would still be orthogonal to the role of positional encoding. Positional encodings exist to reintroduce topology into a permutation-equivariant operator, biasing attention by anchoring content to spatial structure. This mechanism does not require, benefit from, or even meaningfully interact with lossless fusion in an information-theoretic sense, as far as this reviewer is aware. I encourage the authors to *clearly disambiguate the role of information preservation in mixed domain positional embedding*, particularly considering the setting where these are learned by the network.

---

> > ### Author Response · Authors · 2026-02-03
> > **Response to Critique on Numerical Stability and Motivation**
> >
> > We appreciate the reviewer's deep theoretical analysis. You are correct that our discussion of 'injectivity' conflated numerical stability (reconstruction) with representational separability (discrimination). We are grateful for the chance to disambiguate our specific objective.
> >
> > **1. Numerical Stability & Condition Numbers**
> >
> > The reviewer argues that even if the map is analytically injective, the condition number (driven by LeakyReLU slopes and projection matrices) makes the inverse ill-posed and "functionally indistinguishable from a lossy one." We fully agree with the reviewer’s numerical analysis. A LeakyReLU with negative slope $\alpha=0.01$ implies a reverse gain of $100$, and stacked layers indeed exacerbate the condition number.
> >
> > However, we respectfully clarify that our goal is not *Generative Reconstruction* (i.e., we are not building a Normalizing Flow or VAE where stable inversion is required to generate data). Our goal is **Discriminative Representation** for downstream tasks.
> > An ill-conditioned map is only "vacuous" if the goal is to recover inputs $x$ from outputs $y$. For a discriminative model, the requirement is simply that distinct input states $(x, p)$ map to distinct keys $(k)$ in the forward pass. Even if the map stretches the space unevenly (high condition number), it remains **topologically valid** for separating classes.
> >
> > We will revise the text to clarify that "Information Preservation" in our context refers to **"Forward Collision Avoidance"**, ensuring the architecture does not structurally force feature collapse (as bottleneck projections would), rather than guaranteeing a numerically stable inverse for reconstruction.
> >
> > **2. Motivation: Why is Injectivity required for PEs?**
> >
> > The reviewer argues that injectivity is "orthogonal" to positional encoding, as PEs simply need to bias attention or reintroduce topology. We believe injectivity is fundamental, not orthogonal, to the *faithful* reintroduction of topology, especially in fusion.
> > Standard Transformers use additive PEs ($x + p$). In this setting, it is theoretically possible for content $x$ and position $p$ to destructively interfere (collide). If a specific texture pattern plus a specific position sums to a vector identical to a different pair, the topology is *lost*, the network cannot distinguish them.
> >
> > By concatenating and projecting ($Concat(x, p) \to MLP$), we structurally guarantee that the mixing mechanism has the *capacity* to maintain both content and position fidelity without destructive interference. The "Injectivity" proof is the formal guarantee that our architecture prevents this ambiguity. It ensures the network **can** distinguish "Texture A at Position X" from "Texture B at Position Y," a property that is critical for fine-grained tactile tasks but not guaranteed by standard additive methods.
> >
> > **3. Training Dynamics**
> >
> > The reviewer notes that we have no constraints to enforce full rank during training. We agree that this is an empirical observation rather than a constrained optimization.
> >
> > We will soften the language in the final manuscript. We will state that our architecture provides the **structural capacity** for losslessness (avoiding the rank deficiencies of bottleneck methods), and that we **empirically observe** this capacity is maintained during training (Figure 5), rather than claiming a theoretical guarantee of learning dynamics.

---

> ### Comment · Reviewer_JpuX · 2026-02-03
> **New inconsistencies need clarification**
>
> Thanks for the reply.
>
> The paper is very clear that the goal is not reconstruction, but this reviewer (nor other reviewers) never claimed that this was the case. If you invoke information preservation as a guarantee, then numerical conditioning matters, because near-collisions are indistinguishable from loss for downstream discrimination as well. In particular, if specific subspaces are compressed to a high enough degree, it doesn't matter if one has analytic injectivity, the information is gone, and that matters for the attention matrix as well. The inverse is ill-posed because the information is **lost**, so the central claim is simply wrong.
>
> > We believe injectivity is fundamental, not orthogonal, to the faithful reintroduction of topology
>
> Well, the theory does not show this, nor why it should be the case. But there are now issues that leaves this reviewer very confused.
>
> This new argumentation ($x + p$) seems subtly shifted from the claims in the paper; the current manuscript did not invoke a **non-additive positional encoding**. According to Eqs. (2,3,4), the PEs is still additive. Instead, the "information preserving mixing" via the nonlinearity $g$ happens between positions and signals from visual ($V$) and tactile ($T$) sources, which are concatenated, then mixed at a token-embedding level. But at this point, **the argument seems to be made on position and signal**, which seems **wholly inconsistent** with the original presentation in the paper.
>
> As I read Eqs. (2,3,4), there is no mention of trying to fundamentally solve additive positional embedding and "destructive interference" here. Your Eq. (3) literally shows:
>
> $$
> X_{projected} = g([X_{modal}^{visual} ; X_{modal}^{tactile}]).
> $$
>
> This does not imply that you are trying to solve additive mixing at all, since from Eq (2) you literally use
>
> $$
> X_{modal} = X_{content} + PE_{local}
> $$
>
> for either $V$ or $T$. This seems to be precisely the additive PE you are now claiming the MLP $g$ is solving. However, since $g$ operates on the output of the addition ($X_{modal}$), it is mathematically impossible for $g$ to resolve a 'destructive interference' that occurred in the previous step. If $x + p$ collides with $x' + p'$ in Eq. (2), $g$ receives identical inputs in Eq. (3) and must produce identical outputs. Thus, your claimed structural guarantee against additive collision does not exist, as far as this reviewer understands the updated claim.
>
> Before we continue, this new pivot on the role of $g$ is highly confusing, and requires disambiguation by the authors. *I would appriciate a clarification on the exact role of $g$, and the apparent inconsistency between your rebuttal and the method outlined in the paper*.

---

> > ### Author Response · Authors · 2026-02-06
> > **Method Specification: Resolving the Eq.(2–4) vs. Rebuttal Ambiguity (Shapes + Role of $g$)**
> >
> > We thank Reviewer JpuX for the continued careful scrutiny.
> > Your last message correctly identifies an important ambiguity/mismatch between (i) parts of our rebuttal wording that could be read as claiming “collision avoidance / destructive interference resolution,” and (ii) what the submitted manuscript specifies in Eqs. (2–4), where positional encodings are additive before the projection head. We want to be explicit that we did not intend to claim that a downstream token-wise map $g$ can undo collisions created by the additive step $x+\mathrm{PE}$. As you correctly point out, if a collision occurs at that stage, any subsequent token-wise mapping receives identical inputs and cannot recover the distinction. Below we therefore (A) make the implemented method fully explicit and align the manuscript to it, and (B) remove the misleading “collision-avoidance / minimal information loss” framing, keeping only a narrow, testable mechanism statement consistent with the equations and implementation.
> >
> > ### Accountability and scope
> >
> > As written, the submitted draft can be read as implying **feature-axis concatenation** and a projection head of the form $g:\mathbb{R}^{2D}\to\mathbb{R}^{D}$ (e.g., Eq. (3) and the Architecture/Appendix description that specifies an MLP starting with `Linear(2*D, H)`), which would suggest **pre-attention cross-modal mixing inside $g$**. This presentation is not internally consistent across sections, and we take responsibility for the resulting confusion.
> >
> > In the revision, we will make the fusion axis, shapes, and the role of $g$ completely explicit, and ensure that every methodological and theoretical statement matches that specification. **All reported results use token-axis concatenation (sequence concatenation), not feature-axis concatenation.** In all reported experiments, **$g:\mathbb{R}^{D}\to\mathbb{R}^{D}$ is applied token-wise with shared weights.**
> >
> > ### Implementation–manuscript alignment (exactly what was run, with shapes)
> >
> > All reported results were obtained with the following high-level structure:
> >
> > 1. **Local (modality-specific) PEs** added within each modality (additive).
> > 2. **Token-axis concatenation** of visual and tactile token sequences.
> > 3. A **shared token-wise MLP** applied per token (shared weights).
> > 4. A **global PE** added on the concatenated sequence before the transformer.
> >
> > To remove ambiguity, we will include explicit pseudocode + shapes in the revision’s appendix and rewrite Eq. (3) to **explicitly specify token-axis concatenation**, updating the architecture description and appendix accordingly:
> >
> > - $X_v \in \mathbb{R}^{N_v \times D}$, $X_t \in \mathbb{R}^{N_t \times D}$.
> > - $X = \mathrm{concat\_tokens}(X_v + \mathrm{PE}_v,\; X_t + \mathrm{PE}_t) \in \mathbb{R}^{(N_v + N_t) \times D}$.
> > - $X' = g(X)$ (applied token-wise; $g:\mathbb{R}^{D}\to\mathbb{R}^{D}$ with shared weights).
> > - $X'' = X' + \mathrm{PE}_{\mathrm{global}}$.
> >
> > **Important clarification (role of $g$):** In the implementation used in all reported experiments, $g$ is **token-wise** (applied independently to each token with shared weights). Thus, **$g$ does not itself perform cross-modal fusion**; cross-modal interaction occurs in **self-attention over the concatenated token sequence**.
> >
> > **Consequence for theory/claims.** Accordingly, any theoretical discussion that relied on a feature-axis $2D \to D$ projection (or “injective projection” language in that sense) is removed or rewritten to match the token-wise $D \to\ D$ stem used in all reported experiments.
> >
> > Concrete manuscript actions (revision):
> >
> > - Rewrite Eq. (3) with explicit shapes and an explicit $\mathrm{concat\_tokens}(\cdot,\cdot)$ operator.
> > - Make Appendix/Architecture consistent with Eq. (3) (the current text uses `Linear(2*D, H)`, which implies feature-axis concatenation).
> > - Remove/replace any sentences that could be read as “$g$ performs fusion across modalities.”

---

> > ### Author Response · Authors · 2026-02-06
> > **Theoretical Scope Correction: Retractions, Stability Boundaries, and Theorem Re-scoping**
> >
> > ### (1) Additive-PE collisions: explicit logic + retraction of prior framing
> >
> > In the submitted manuscript, local PEs are additive (Eq. (2)):
> >
> > $$
> > X^{\mathrm{modal}} = X + \mathrm{PE}.
> > $$
> >
> > If two distinct pairs collide at this additive stage, i.e.
> >
> > $$
> > x_i + \mathrm{pe}_i = x_j + \mathrm{pe}_j,
> > $$
> >
> > then **any downstream computation that depends only on the collided token representation cannot recover the distinction.** In particular, a downstream token-wise stem $g$ cannot “undo” such collisions.
> >
> > Accordingly, we explicitly retract and remove any framing that could be read as claiming **collision avoidance**, **destructive interference resolution**, **minimal information loss**, or **information preservation** as a guaranteed property of our method.
> >
> > ### (2) Boundary on numerical stability / inversion
> >
> > We also add an explicit boundary aligned with your numerical stability critique:
> >
> > **We make no claim about stable invertibility or well-conditioned inversion of any learned map after training, and we removed any text that could be read as implying such a guarantee.**
> >
> > ### (3) Theory cleanup to be unambiguous, minimal, and consistent with the implemented model
> >
> > - **Prop. 3.3:** removed. We delete the associated entropy/volume-preservation language entirely.
> > - **Thm. 3.1:** in the revision, this is no longer treated as a main pillar. We will **rename it from “Theorem” to a “Remark/Observation”** and move it to the appendix. It will be stated only as an architectural non-bottleneck observation under explicit full-rank assumptions, **not** as a learning guarantee and **not** used to motivate performance.
> > - **Thm. 3.2:** renamed to token re-indexing / permutation consistency, explicitly **not** translation equivariance of the full PE-conditioned transformer. We include it only to ensure that injecting positional signals within the token-wise stem does not introduce unintended order dependence beyond the explicit PE indexing.
> >
> > ### (4) Global scrubbing for consistency
> >
> > To prevent leftover overclaiming, we will scrub the manuscript for “provable guarantees” phrasing and remove it from the abstract, introduction, conclusion, and any figure/table captions where it appears.

---

> ### Author Response · Authors · 2026-02-06
> **Empirical Evidence Clarification: Ablations, Controls, and Transfer Terminology**
>
> We report mean ± std over 5 seeds. All variants are trained/evaluated under the same protocol and compute budgets (same epochs, optimizer/LR schedule, batch size, and augmentation); only the stated architectural component differs.
>
> **Definitions (variants).**
> - **ViTaPEs:** local PE before nonlinear token-wise $g$.
> - **PE-after-$g$:** same, but local PE added after $g$.
> - **Linearized-$g$:** same 2-layer token-wise MLP, activation replaced by identity.
> - **$g=\mathrm{Id}$:** remove the stem (identity mapping).
>
> **Parameter-matched nonlinearity control.** Linearized-$g$ differs only by removing the nonlinearity; all weights remain learned.
>
> $$
> g(x) = W_2\bigl(W_1 x + b_1\bigr) + b_2
> $$
>
> | Variant | TAG Category (Top-1 %) | TAG→OF-Real Linear Probe (Top-1 %) | TAG→OF-Real Zero-shot (Top-1 %) |
> |---|---:|---:|---:|
> | ViTaPEs | 75.9 ± 0.2 | 68.1 ± 0.3 | 65.2 ± 2.0 |
> | PE-after-$g$ | 71.3 ± 0.4 | 56.2 ± 0.6 | 45.7 ± 2.0 |
> | Linearized-$g$ ($\sigma=\mathrm{Id}$) | 72.4 ± 0.2 | 54.7 ± 1.7 | 51.4 ± 1.5 |
> | $g=\mathrm{Id}$ | 70.9 ± 0.1 | 52.9 ± 0.4 | 48.1 ± 1.1 |
>
> **Interpretation:** performance is best when (i) $g$ is nonlinear and (ii) the local PE is injected before $g$, with larger deltas under out-of-domain transfer. Since Linearized-$g$ matches the nonlinear $g$ in parameterization, the ViTaPEs–Linearized-$g$ gap isolates the effect of nonlinearity/conditioning rather than added capacity.
>
> **What this table does *not* claim.** These ablations do not support any claim of collision avoidance, information preservation, or stable inversion; they only support the conditioning/placement mechanism.
>
> **Terminology.** TAG→OF-Real is **cross-dataset transfer** and, in our setup, it also induces a **sensor shift**,  i.e., a compound OOD shift. We reserve “cross-sensor transfer” for experiments designed to isolate hardware effects under a matched protocol.

---

> ### Author Response · Authors · 2026-02-13
> **Theoretical Re-scoping, Methodological Precision, and Novel Ablations**
>
> We have uploaded the revised manuscript. We want to thank you again for your rigorous theoretical scrutiny; your feedback was instrumental in helping us pivot the paper away from mathematically imprecise claims and toward the true strength of our work.
>
> As committed during the discussion phase, we have made the following concrete changes in the newly revised manuscript:
>
> 1. **Theoretical Claims Removed & Re-scoped (Sections 1, 3, & Appendix B):** We have entirely removed the claims regarding "provable guarantees," "minimal information loss," and "volume preservation" from the Abstract, Introduction, and Conclusion. Proposition 3.3 has been completely deleted. Theorem 3.2 has been appropriately downgraded to "Token re-indexing consistency" and moved to Appendix B (Proposition B.1), ensuring we no longer falsely claim spatial translation equivariance.
> 2. **Explicit Equations and Tensor Shapes (Section 3.2):** We rewrote Equations 3 and 4 to explicitly specify *token-axis concatenation* ($X_{concat} \in \mathbb{R}^{N \times D}$) and explicitly stated that the non-linear stem $g$ is applied *token-wise* (row-wise) with shared weights. This resolves the logical inconsistency regarding "additive collisions" that you correctly pointed out in the previous draft.
> 3. **New Projection Head ($g$) Ablation (Section 5, Table 5):** To address your concern that the performance gains might simply be due to the general capacity of the Transformer rather than our specific geometry handling, we added a new ablation study (Table 5). This includes the parameter-matched `Linearized-g` control and the `PE-after-g` variant. The results empirically isolate and validate that injecting spatial structure *before* a token-wise non-linearity is exactly what drives the out-of-domain transfer performance, effectively proving the structural value of our architecture.
> 4. **Terminology Adjustments:** We have replaced "Novel Framework" with "transformer-based architecture" throughout the text to avoid overclaiming.
>
> We believe these revisions fully align the manuscript with the empirical reality of the method, and we thank you for guiding us to a much stronger, more precise paper.

---

### Decision · Action_Editor_FRMV · 2026-04-03

**Recommendation:** Accept with minor revision

**Audience:**

Yes

**Audience Explanation:**

While the scope of this work is rather specific, on multi-modality in the form of visuo-tactile modeling for robotics task, the more general topic of multi-modality has a broad audience. The same is true for variations of transformer architectures and position encodings which currently enjoy nearly ubiquitous application to many kinds of data. Furthermore learning for robotics is in scope for TMLR as are other topics on learning for specific disciplines.

All reviewers agreed that there is an audience.

**Claims And Evidence:**

Yes

**Claims Explanation:**

The main claims of the work following revision are that (1) a particular composition of local, per-modality position encodings followed by global, cross-modality position encodings is apt and novel for visuo-tactile modeling and (2) empirically this approach to position encoding improves results on recognition tasks (like material and object classification) and recognition for control (robotic grasping success/failure prediction).

These claims are supported by evidence by (1) the discussion of related work on foundation models and VTT and (2) the experimental results (Fig. 1, Tab. 1, 2, 3) and further analysis and ablations.

2/3 reviewers are satisfied with the claims and evidence but one reviewer disagrees and argues the bar is not met for lack of methodological generality, a new principle, or strong empirical surprise. The action editor sides with satisfaction of the claims and evidence: the work is informative and provides what it promises to deliver.

Note: The work has changed substantially during discussion and revision and so the decision is a function of the final state and not earlier theoretical claims and material that have since been revised and removed. Due to the amount of revision, the decision is to accept with a final minor revision, so that the authors can fully incorporate the feedback from review.

---

> ### Author Response · Authors · 2026-04-17
> **Final Minor Revision Submitted**
>
> Dear Action Editor,
>
> Thank you very much for your thoughtful decision and for handling the review process so carefully.
>
> We appreciate your assessment of the final revised manuscript and your recognition that the paper’s claims are supported in its revised form. We are also grateful for the constructive feedback from you and the reviewers, which helped us clarify the scope, presentation, and positioning of the work.
>
> We have now uploaded the revised camera-ready manuscript incorporating all requested minor revisions from the reviewers.
>
> Thank you again for your time, consideration, and guidance throughout the process.
>
> Best regards,
> the Authors